# Multi-Agent Off-Policy TDC with Near-Optimal Sample and Communication Complexities

**Ziyi Chen**  *u1276972@utha.edu*
*Department of Electrical and Computer Engineering*
*University of Utah*

**Yi Zhou**  *yi.zhou@utah.edu*
*Department of Electrical and Computer Engineering*
*University of Utah*

**Rong-Rong Chen**  *rchen@ece.utah.edu*
*Department of Electrical and Computer Engineering*
*University of Utah*

**Reviewed on OpenReview:** *https://openreview.net/forum?id=tnPjQpYk7D*

## Abstract

The finite-time convergence of off-policy temporal difference (TD) learning has been comprehensively studied recently. However, such a type of convergence has not been established for off-policy TD learning in the multi-agent setting, which covers broader reinforcement learning applications and is fundamentally more challenging. This work develops a decentralized TD with correction (TDC) algorithm for multi-agent off-policy TD learning under Markovian sampling. In particular, our algorithm avoids sharing the actions, policies and rewards of the agents, and adopts mini-batch sampling to reduce the sampling variance and communication frequency. Under Markovian sampling and linear function approximation, we proved that the finite-time sample complexity of our algorithm for achieving an $\epsilon$-accurate solution is in the order of $\mathcal{O}\big(\frac{M \ln \epsilon^{-1}}{\epsilon (1-\sigma_2)^2}\big)$, where $M$ denotes the total number of agents and $\sigma_2$ is a network parameter. This matches the sample complexity of the centralized TDC. Moreover, our algorithm achieves the optimal communication complexity $\mathcal{O}\big(\frac{\sqrt{M} \ln \epsilon^{-1}}{1-\sigma_2}\big)$ for synchronizing the value function parameters, which is order-wise lower than the communication complexity of the existing decentralized TD(0). Numerical simulations corroborate our theoretical findings.

## 1 Introduction

Multi-agent reinforcement learning (MARL) is an emerging technique that has broad applications in control Yanmaz et al. (2017); Chalaki & Malikopoulos (2020), wireless sensor networks Krishnamurthy et al. (2008); Yuan et al. (2020), robotics Yan et al. (2013), etc. In MARL, agents cooperatively interact with an environment and follow their own policies to collect local rewards. In particular, policy evaluation is a fundamental problem in MARL that aims to learn a multi-agent value function associated with the policies of the agents. This motivates the development of convergent and communication-efficient multi-agent TD learning algorithms.

For single-agent on-policy evaluation (i.e., samples are collected by target policy), the conventional TD(0) algorithm Sutton (1988); Sutton & Barto (2018) and Q-learning algorithm Dayan (1992) have been developed with asymptotic convergence guarantee. Recently, their finite-time (i.e., non-asymptotic) convergence has been established under Markovian sampling (i.e., the samples are obtained from Markovian decision process and thus are not i.i.d.) and linear approximation Bhandari et al. (2018); Zou et al. (2019). However, these algorithms may diverge in the off-policy setting Baird (1995), where samples are collected by a different

behavior policy. To address this important issue, a family of gradient-based TD (GTD) algorithms were developed for off-policy evaluation with asymptotic convergence guarantee Sutton et al. (2008; 2009); Maei (2011). In particular, the TD with gradient correction (TDC) algorithm has been shown to have superior performance and its finite-time convergence has been established recently under Markovian sampling Xu et al. (2019); Gupta et al. (2019); Kaledin et al. (2020).

For multi-agent on-policy evaluation, various decentralized TD learning algorithms have been developed. For example, the finite-time convergence of decentralized TD(0) was established with i.i.d samples Wai et al. (2018); Doan et al. (2019) and Markovian samples Sun et al. (2020), respectively, under linear function approximation, and an improved result is further obtained in Wang et al. (2020) by leveraging gradient tracking. However, these algorithms do not apply to the off-policy setting. In the existing literature, decentralized off-policy TD learning has been studied only in simplified settings, for example, agents obtain independent MDP trajectories Macua et al. (2014); Stanković & Stanković (2016); Cassano et al. (2020) or share their behavior and target policies with each other Cassano et al. (2020), and the data samples are either i.i.d. or have a finite sample size. These MARL settings either are impractical or reveal the agents' policies that may be sensitive. Therefore, we want to ask the following question:

- *Q1: Can we develop a decentralized off-policy TD algorithm for MARL with interdependent agents and avoids sharing local actions, policies and rewards of the agents?*

In fact, developing such a desired decentralized off-policy TD learning algorithm requires overcoming two major challenges. First, to perform decentralized off-policy TD learning, all the agents need to obtain a global importance sampling ratio (see Section 3.2). In Cassano et al. (2020), the authors obtained this ratio by sharing all local policies among the agents, which may lead to information leakage. Therefore, we aim to develop a safer scheme to synchronize the global importance sampling ratio among the agents without sharing any sensitive local information. Second, the existing decentralized TD-type algorithm achieves a communication complexity (number of communication rounds) of $\mathcal{O}\big(\big(\epsilon^{-1} + \frac{\sqrt{M}}{\sqrt{\epsilon}(1-\sigma_2)}\big)\ln\epsilon^{-1}\big)$ for networks with $M$ agents and parameter $\sigma_2$ (See Assumption 5 in Section 4 for the definition of $\sigma_2$) Sun et al. (2020). This induces much communication overhead when the target accuracy $\epsilon$ is small. Hence, we want to ask the following theoretical question:

- *Q2: Can we develop a decentralized off-policy TD learning algorithm that achieves a near-optimal finite-time sample complexity and a near-optimal communication complexity under Markovian sampling?*

In this work, we provide affirmative answers to these questions by developing a decentralized TDC algorithm that avoids sharing the sensitive information of the agents and achieves the near-optimal sample complexity as well as a significantly reduced communication complexity. We summarize our contributions as follows.

### 1.1 Summary of Contribution

To perform multi-agent off-policy evaluation, we develop a decentralized TDC algorithm with linear function approximation. In every iteration, agents perform two-timescale TDC updates locally and exchange model parameters with their neighbors. In particular, our algorithm adopts the following designs to avoid sharing agents' local sensitive information and reduce communication load.

- We let the agents perform local averaging on their local importance sampling ratios to obtain approximated global importance sampling ratios.

- All the agents use a mini-batch of samples to update their model parameters in each iteration. The mini-batch sampling reduces the sampling variance and the communication frequency, leading to an improved communication complexity over that of the existing decentralized TD(0).

- After the decentralized TDC iterations, our algorithm performs additional local averaging steps to achieve a global consensus on the model parameters. This turns out to be critical for achieving the near-optimal complexity bounds.

Theoretically, we analyze the finite-time convergence of this decentralized TDC algorithm with Markovian samples and show that it attains a fast linear convergence. The overall sample complexity for achieving

an $\epsilon$-accurate solution is in the order of $\mathcal{O}\left(\frac{M\ln\epsilon^{-1}}{\epsilon(1-\sigma_2)^2}\right)$. When there is a single agent $(M=1)$, this sample complexity result matches that of centralized TDC Xu et al. (2019) and matches the theoretical lower bound $\mathcal{O}(\epsilon^{-1})$ Kaledin et al. (2020) up to a logarithm factor. In addition, the sample complexity is proportional to $M$, which matches the theoretical lower bound of decentralized strongly convex optimization Scaman et al. (2017). Moreover, the communication complexity of our algorithm for synchronizing value function parameters is in the order of $\mathcal{O}\left(\frac{\sqrt{M}\ln\epsilon^{-1}}{1-\sigma_2}\right)$, which is significantly lower than the communication complexity $\mathcal{O}\left(\epsilon^{-1}+\frac{\sqrt{M}}{\sqrt{\epsilon}(1-\sigma_2)}\ln\epsilon^{-1}\right)$ of the decentralized TD(0) Sun et al. (2020) and matches the communication complexity lower bound Scaman et al. (2017).

Technically, our analysis is a nontrivial generalization of the analysis of centralized off-policy TDC to the decentralized case. In particular, our analysis establishes tight bounds of the consensus error caused by synchronizing the global importance sampling ratio, especially under the Markovian sampling where the data samples are correlated. Moreover, we strategically bound the estimation error of the global importance sampling logarithm-ratio. Please refer to the proof sketch at the end of Section 4 for details.

## 1.2 Other Related Work

**Centralized policy evaluation.** TD(0) with linear function approximation Sutton (1988) is popular for on-policy evaluation. The asymptotic and non-asymptotic convergence results of TD(0) have been established in Sutton (1988); Dayan (1992); Jaakkola et al. (1993); Gordon (1995); Baird (1995); Tsitsiklis & Van Roy (1997); Tadić (2001); Hu & Syed (2019) and Korda & La (2015); Liu et al. (2015); Bhandari et al. (2018); Dalal et al. (2018b); Lakshminarayanan & Szepesvari (2018); Wang et al. (2019); Srikant & Ying (2019); Xu et al. (2020b) respectively. Sutton et al. (2009) proposed TDC for off-policy evaluation. The finite-sample convergence of TDC has been established in Dalal et al. (2018a; 2020) with i.i.d. samples and in Xu et al. (2019); Gupta et al. (2019); Kaledin et al. (2020) with Markovian samples.

**Decentralized policy evaluation.** Mathkar & Borkar (2016) proposed the decentralized TD(0) algorithm. The asymptotic and non-asymptotic convergence rate of decentralized TD have been obtained in Borkar (2009) and Sun et al. (2020); Wang et al. (2020) respectively. Exisitng decentralized off-policy evaluation studies considered simplified settings. Macua et al. (2014); Stanković & Stanković (2016) obtained asymptotic result for decentralized off-policy evaluation where the agents obtained independent MDPs. Cassano et al. (2020) obtained linear convergence rate also with independent MDPs by applying variance reduction and extended to the case where the individual behavior policies and the joint target policy are shared among the agents.

**Decentralized policy control.** Decentralized policy control is also an important MARL problem where the goal is to learn the optimal policy for each agent. Many algorithms have been proposed for decentralized policy control, including policy gradient Chen et al. (2021) and actor-critic Qu et al. (2020); Zhang et al. (2021).

## 2 Policy Evaluation in Multi-Agent RL

In this section, we introduce multi-agent reinforcement learning (MARL) and define the policy evaluation problem. Consider a fully decentralized multi-agent network that consists of $M$ agents. The network topology is specified by an undirected graph $\mathcal{G}=(\mathcal{M},\mathcal{E})$, where $\mathcal{M}=\{1,2,\cdots,M\}$ denotes the set of agents and $\mathcal{E}$ denotes the set of communication links. In other words, each agent $m$ only communicates with its neighborhood $\mathcal{N}_m := \{m'\in\mathcal{M}:(m,m')\in\mathcal{E}\}$. In MARL, the agents interact with a dynamic environment through a multi-agent Markov decision process (MMDP) specified as $\{\mathcal{S},\{\mathcal{A}^{(m)}\}_{m=1}^M,P,\{R^{(m)}\}_{m=1}^M,\gamma\}$. To elaborate, $\mathcal{S}$ denotes a global state space that is shared by all the agents, $\mathcal{A}^{(m)}$ corresponds to the action space of agent $m$, $P$ is the state transition kernel and $R^{(m)}$ denotes the reward function of agent $m$. All the state and action spaces have finite cardinality. $\gamma\in(0,1]$ is a discount factor.

At any time $t$, assume that all the agents are in the global state $s_t\in\mathcal{S}$. Then, each agent $m$ takes a certain action $a_t^{(m)}\in\mathcal{A}^{(m)}$ following its own stationary policy $\pi^{(m)}$, i.e., $a_t^{(m)}\sim\pi^{(m)}(\cdot|s_t)$. After all the actions are taken, the global state transfers to a new state $s_{t+1}$ according to the transition kernel $P$, i.e., $s_{t+1}\sim P(\cdot|s_t,a_t)$

where $a_t := \{a_t^{(m)}\}_{m=1}^M$. At the same time, each agent $m$ receives a local reward $R_t^{(m)} := R^{(m)}(s_t, a_t, s_{t+1})$ from the environment for this action-state transition. Throughout the MMDP, each agent $m$ has access to only the global state $\{s_t\}_t$ and its own actions $\{a_t^{(m)}\}_t$ and rewards $\{R_t^{(m)}\}_t$. The goal of policy evaluation in MARL is to evaluate the following value function associated with all the local policies $\pi := \{\pi^{(m)}\}_{m=1}^M$ for any global state $s$.

$$V^\pi(s) = \mathbb{E}\Big[\sum_{t=0}^{+\infty} \gamma^t \Big(\frac{1}{M}\sum_{m=1}^M R_t^{(m)}\Big)\Big|s_0 = s, \pi\Big]. \tag{1}$$

A popular algorithm for policy evaluation in MARL is the decentralized TD(0) Sun et al. (2020). Specifically, consider a popular linear function approximation of the value function $V_\theta(s) := \theta^\top \phi(s)$, where $\theta \in \mathbb{R}^d$ contains the model parameters and $\phi(s)$ is a feature vector that corresponds to the state $s$. In decentralized TD(0), each agent $m$ collects a single Markovian sample $\{s_t, a_t^{(m)}, s_{t+1}, R_t^{(m)}\}$ at time $t$ $(a_t^{(m)} \sim \pi^{(m)}(\cdot|s_t)$, $s_{t+1} \sim P(\cdot|s_t, a_t)$, $R_t^{(m)} := R^{(m)}(s_t, a_t, s_{t+1}))$ and updates its own model parameters $\theta_t^{(m)}$ with learning rate $\alpha > 0$ as follows.

$$\theta_{t+1}^{(m)} = \sum_{m' \in \mathcal{N}_m} U_{m,m'}\theta_t^{(m')} + \alpha\big(A_t\theta_t^{(m)} + b_t^{(m)}\big), \tag{2}$$

where $U$ corresponds to a doubly stochastic communication matrix, each agent $m$ only communicates with its neighborhood $\mathcal{N}_m$, and $A_t = \phi(s_t)(\gamma\phi(s_{t+1}) - \phi(s_t))^\top$, $b_t^{(m)} = R_t^{(m)}\phi(s_t)$. The above update rule applies the local TD error to update the parameters and synchronize the parameters among neighboring agents through the network. It can be inferred that $\theta_t^{(m)}$ obtained by this decentralized TD(0) algorithm is $\epsilon$-close to the optimal solution with both sample complexity (the number of required samples) and communication complexity (the number of communication rounds) being $\mathcal{O}(\epsilon^{-1} + \sqrt{M/\epsilon}(1 - \sigma_2)^{-1})\ln(\epsilon^{-1})$. [1]

## 3 Two-Timescale Decentralized TDC for Off-Policy Evaluation

### 3.1 Centralized TDC

In this subsection, we review the centralized TD with gradient correction (TDC) algorithm Sutton et al. (2009). In RL, the agent may not have enough samples that are collected following the target policy $\pi$. Instead, it may have some data samples that are collected under a different behavior policy $\pi_b$. Therefore, in this *off-policy* setting, the agent would like to utilize the historical data to help evaluate the value function $V^\pi$ associated with the target policy $\pi$.

In Sutton et al. (2009), a family of gradient-based TD (GTD) learning algorithms have been proposed for off-policy evaluation. In particular, the TDC algorithm has been shown to have superior performance. To explain, consider the linear approximation $V_\theta(s) = \theta^\top\phi(s)$ and suppose the state space includes states $s_1, ..., s_n$, we can define a total value function $V_\theta := [V_\theta(s_1), ..., V_\theta(s_n)]^\top$. The goal of TDC is to minimize the following mean square projected Bellman error (MSPBE).

$$\text{MSPBE}(\theta) := \mathbb{E}_{\mu_b}\|V_\theta - \Pi T^\pi V_\theta\|^2,$$

where $\mu_b$ is the stationary distribution under the behavior policy $\pi_b$, $T^\pi$ is the Bellman operator and $\Pi(V) := \arg\min_{V_\theta}\|V_\theta - V\|^2$ is a projection operator of any state value function $V : \mathcal{S} \to \mathbb{R}$ onto the space of linear value functions $\{V_\theta : V_\theta(s) = \theta^\top\phi(s)\}$. Given the $i$-th sample $(s_i, a_i, s_{i+1}, R_i)$ obtained by the behavior policy, we define the following terms

$$\rho_i := \frac{\pi(a_i|s_i)}{\pi_b(a_i|s_i)}, \quad b_i := \rho_i R_i\phi(s_i), \quad A_i := \rho_i\phi(s_i)(\gamma\phi(s_{i+1}) - \phi(s_i))^\top,$$
$$B_i := -\gamma\rho_i\phi(s_{i+1})\phi(s_i)^\top, \quad C_i := -\phi(s_i)\phi(s_i)^\top, \tag{3}$$

---

[1] Sun et al. (2020) does not report sample complexity and communication complexity, so we calculated them based on their finite-time error bound in proposition 2.

where $\rho_i$ is referred to as the *importance sampling ratio*. Then, with learning rates $\alpha, \beta > 0$ and initialization parameters $\theta_0, w_0$, the two timescale off-policy TDC algorithm takes the following recursive updates for iterations $t = 0, 1, 2, ...$

$$\text{(TDC):} \quad \begin{cases} \theta_{t+1} = \theta_t + \alpha(A_t \theta_t + b_t + B_t w_t), \\ w_{t+1} = w_t + \beta(A_t \theta_t + b_t + C_t w_t). \end{cases} \tag{4}$$

Xu et al. (2019); Xu & Liang (2020) study slight variations of the above TDC algorithm by using projection and minibatch technique respectively, and obtain that both variants obtain an $\epsilon$-approximation of the optimal model parameter $\theta^* = -A^{-1}b$ ($A := \mathbb{E}_{\pi_b}[A_i]$, $b := \mathbb{E}_{\pi_b}[b_i]$) with sample complexity $\mathcal{O}(\epsilon^{-1} \ln \epsilon^{-1})$.

## 3.2 Decentralized Mini-batch TDC

In this subsection, we propose a decentralized TDC algorithm for off-policy evaluation in MARL. In the multi-agent setting, without loss of generality, we assume that each agent $m$ has a target policy $\pi^{(m)}$ and its samples are collected by a different behavior policy $\pi_b^{(m)}$. In particular, if agent $m$ is on-policy, then we have $\pi_b^{(m)} = \pi^{(m)}$. In this *multi-agent off-policy* setting, the agents aim to utilize the data collected by the behavior policies $\pi_b = \{\pi_b^{(m)}\}_{m=1}^M$ to help evaluate the value function $V^\pi$ associated with the target policies $\pi = \{\pi^{(m)}\}_{m=1}^M$.

However, directly generalizing the centralized TDC algorithm to the decentralized setting will encounter several challenges. First, the centralized TDC in eq. (4) consumes one sample per-iteration and achieves the sample complexity $O(\epsilon^{-1} \log \epsilon^{-1})$ Xu et al. (2019). Therefore, the corresponding decentralized TDC would perform one local communication per-iteration and is expected to have a communication complexity in the order of $O(\epsilon^{-1} \log \epsilon^{-1})$, which induces *large communication overhead*. Second, in the multi-agent off-policy setting, every agent $m$ has a local importance sampling ratio $\rho_i^{(m)} := \pi^{(m)}(a_i^{(m)}|s_i)/\pi_b^{(m)}(a_i^{(m)}|s_i)$. However, to correctly perform off-policy updates, every agent needs to know all the other agents' local importance sampling ratios in order to obtain the **global importance sampling ratio** $\rho_i := \prod_{m=1}^M \rho_i^{(m)}$. To address these challenges that are not seen in decentralized TD(0) and centralized TDC , we next propose a decentralized TDC algorithm that takes mini-batch stochastic updates.

To elaborate, note that $\rho_i$ can be rewritten as

$$\rho_i = \exp\left(M \cdot \tfrac{1}{M} \sum_{m=1}^M \ln \rho_i^{(m)}\right).$$

Therefore, all the agents just need to obtain the average $\frac{1}{M} \sum_{m=1}^M \ln \rho_i^{(m)}$, which can be computed via local communication of the logarithm-ratios $\{\ln \rho_i^{(m)}\}_{m=1}^M$ for $L$ rounds. Specifically, every agent $m$ initializes $\widetilde{\rho}_{i,0}^{(m)} = \ln \rho_i^{(m)}$ and for iterations $\ell = 0, ..., L-1$ do

$$\widetilde{\rho}_{i,\ell+1}^{(m)} = \sum_{m' \in \mathcal{N}_m} U_{m,m'} \widetilde{\rho}_{i,\ell}^{(m')}, \tag{5}$$

$$\text{(Output)}: \quad \widehat{\rho}_i^{(m)} = \exp(M \cdot \widetilde{\rho}_{i,L}^{(m)}). \tag{6}$$

In Corollary 2 (see the appendix), we prove that all of these local estimates $\{\widehat{\rho}_i^{(m)}\}_{m=1}^M$ converge exponentially fast to the desired quantity $\rho_i$ as $L$ increases. Then, every agent $m$ performs the following two-timescale TDC updates

$$\theta_{t+1}^{(m)} = \sum_{m' \in \mathcal{N}_m} U_{m,m'} \theta_t^{(m')} + \frac{\alpha}{N} \sum_{i=tN}^{(t+1)N-1} \left(A_i^{(m)} \theta_t^{(m)} + \widehat{b}_i^{(m)} + B_i^{(m)} w_t^{(m)}\right), \tag{7}$$

$$w_{t+1}^{(m)} = \sum_{m' \in \mathcal{N}_m} U_{m,m'} w_t^{(m')} + \frac{\beta}{N} \sum_{i=tN}^{(t+1)N-1} \left(A_i^{(m)} \theta_t^{(m)} + \widehat{b}_i^{(m)} + C_i w_t^{(m)}\right), \tag{8}$$

where $A_i^{(m)}, B_i^{(m)}, \widehat{b}_i^{(m)}$ are defined by replacing the global variables $\rho_i$ and $R_i$ involved in $A_i, B_i, b_i^{(m)}$ (see eq. (3)) with local variables $\widehat{\rho}_i^{(m)}$ and $R_i^{(m)}$ respectively. To summarize, every TDC iteration of Algorithm 1 consumes $N$ Markovian samples, and requires two vector communication rounds for synchronizing the parameter vectors $\theta_t^{(m)}, w_t^{(m)}$, and $L$ scalar communication rounds for estimating the global importance sampling ratio ($\{\rho_{i,\ell}^{(m)} : i = tN, \ldots, (t+1)N - 1, m \in \mathcal{M}\}$ are shared in the $\ell$-th communication round). We summarize these update rules in Algorithm 1. Moreover, after the decentralized TDC updates, the agents perform additional $T'$ local averaging steps to reach a global consensus on the model parameters.

---

**Algorithm 1** Decentralized mini-batch TDC.

---

**Input:** Batch size $N$, iterations $T, T'$, learning rates $\alpha, \beta$.
**Initialize:** $\theta_0^{(m)}, w_0^{(m)}$ for all agents $m \in \mathcal{M}$.
**for** *iteration $t = 0, 1, \ldots, T - 1$* **do**
    Each agent collects $N$ Markovian samples and computes their local importance sampling ratios $\rho_i^{(m)}$.
    **for** *communication round $\ell = 0, 1, \ldots, L - 1$* **do**
        **for** *agent $m \in \mathcal{M}$ in parallel* **do**
            Communicate $\widetilde{\rho}_{i,\ell}^{(m)}$ via eq. (5) for the $N$ samples $i = tN, \ldots, (t+1)N - 1$.
        **end**
    **end**
    **for** *agent $m \in \mathcal{M}$ in parallel* **do**
        Agent $m$ estimates global importance sampling ratios $\widehat{\rho}_i^{(m)}$ via eq. (6) for $i = tN, \ldots, (t+1)N - 1$, and then performs the updates in eqs. (7) and (8).
    **end**
**end**
**for** *iteration $t = T, T + 1, \ldots, T + T' - 1$* **do**
    **for** *agent $m \in \mathcal{M}$ in parallel* **do**
        $\theta_{t+1}^{(m)} = \sum_{m' \in \mathcal{N}_m} U_{m,m'} \theta_t^{(m')}$.
    **end**
**end**
**Output:** $\{\theta_{T+T'}^{(m)}\}_{m=1}^M$.

---

## 4 Finite-Time Analysis of Decentralized TDC

In this section, we analyze the finite-time convergence of Algorithm 1. Denote $\mu_{\pi_b}$ as the stationary distribution of the Markov chain $\{s_t\}_t$ induced by the collection of agents' behavioral policies $\pi_b$. Throughout the analysis, we define the following notations.

$$A := \mathbb{E}_{\pi_b}[A_i], \ B := \mathbb{E}_{\pi_b}[B_i], \ C := \mathbb{E}_{\pi_b}[C_i], \ \overline{b}_i := \frac{1}{M}\sum_{m=1}^M b_i^{(m)}, \ \overline{b} := \mathbb{E}_{\pi_b}[\overline{b}_i],$$
$$\overline{\theta}_t := \frac{1}{M}\sum_{m=1}^M \theta_t^{(m)}, \ \theta^* := -A^{-1}\overline{b}, \ w_t^* := -C^{-1}(A\overline{\theta}_t + \overline{b}),$$

where $\mathbb{E}_{\pi_b}$ denotes the expectation when $s_t \sim \mu_{\pi_b}$, $a_t^{(m)} \sim \pi_b^{(m)}(s_t)$ and $s_{t+1} \sim P(\cdot|s_t, a_t)$, $A_i, B_i, C_i$ are defined in eq. (3) with exact global importance sampling ratio $\rho_i$, $\theta^*$ is the optimal model parameter, and $w_t^*$ is the optimal auxiliary parameter corresponding to $\overline{\theta}_t$ . It is well-known that the optimal model parameter is $\theta^* = -A^{-1}\overline{b}$ Xu et al. (2020b); Xu & Liang (2020). We make the following standard assumptions.

**Assumption 1.** *There exist constants $\nu > 0$ and $\delta \in (0, 1)$ such that for all $t \geq 0$,*

$$\sup_{s \in \mathcal{S}} d_{TV}\left(\mathbb{P}_{\pi_b}\left(s_t \mid s_0 = s\right), \mu_{\pi_b}\right) \leq \nu\delta^t, \tag{9}$$

*where $d_{TV}$ denotes the total-variation distance.*

**Assumption 2.** *The matrices $A$ and $C$ are invertible.*

**Assumption 3.** *The feature vectors satisfy $\|\phi(s)\| \leq 1, \forall s$.*

**Assumption 4.** *There exist $R_{\max}, \rho_{\max} > 0$ such that for all $m \in \mathcal{M}$: $\max_{s,a,s'} R^{(m)}(s, a, s') < R_{\max}$ and $\max_{s,a^{(m)}} \rho^{(m)}(s, a^{(m)}) < \rho_{\max}$ where $\rho^{(m)}(s, a^{(m)}) := \frac{\pi^{(m)}(a^{(m)}|s)}{\pi_b^{(m)}(a^{(m)}|s)}$ denotes the global importance sampling ratio .*

**Assumption 5.** *The communication matrix $U$ is doubly stochastic, (i.e., the entries of each row and those of each column sum up to 1.) , and its second largest singular value satisfies $\sigma_2 \in [0, 1)$.*

Assumption 1 has been widely adopted in the existing literature Bhandari et al. (2018); Xu et al. (2019); Xu & Liang (2020); Shaocong et al. (2020; 2021). It holds for all homogeneous Markov chains with finite state-space and all uniformly ergodic Markov chains. Assumptions $2 - 4$ are widely adopted in the analysis of TD learning algorithms Xu et al. (2019); Xu & Liang (2020). In particular, Assumption 2 implies that $\lambda_1 := -\lambda_{\max}(A^\top C^{-1} A) > 0$, $\lambda_2 := -\lambda_{\max}(C) > 0$ where $\lambda_{\max}(C)$ denotes the largest eigenvalue of matrix $C$. Assumption 3 can always hold by normalizing the feature vectors. Assumption 4 holds for any uniformly lower bounded behavior policy, i.e., $\inf_{(s,a,m)} \pi_b^{(m)}(s, a) > 0$, which ensures that every state-action pair $(s, a)$ is visited infinitely often. Assumption 5 is standard in decentralized optimization Singh et al. (2020); Saha et al. (2020) and TD learning Sun et al. (2020); Wang et al. (2020). $\sigma_2$ is an important measure that reflects communication topology. For example, densely connected network tends to have smaller $\sigma_2$ than sparse network.

We obtain the following finite-time error bound as well as the sample complexity (the number of required samples) and communication complexity (the number of communication rounds) for Algorithm 1 with Markovian samples.

**Theorem 1.** *Let Assumptions 1–5 hold. Run Algorithm 1 for $T$ iterations with learning rates $\alpha \leq \min\{\mathcal{O}(\beta), \mathcal{O}(\frac{1-\sigma_2}{\sqrt{M}})\}$, $\beta \leq \mathcal{O}(1)$, batch size $N \geq \max\{\mathcal{O}(1), \mathcal{O}(\frac{\beta}{\alpha})\}$ and $L \geq \mathcal{O}(\frac{\ln M + M \ln \rho_{\max}}{\ln \sigma_2^{-1}})$ (see eq. (35)-(38) for the full expressions ). Then, we have the following convergence rate (see eq. (28) for its full expression)*

$$\mathbb{E}\big[\|\overline{\theta}_T - \theta^*\|^2\big] \leq \Big(1 - \frac{\alpha \lambda_1}{6}\Big)^T \big(\|\overline{\theta}_0 - \theta^*\|^2 + \|\overline{w}_0 - w_0^*\|^2\big) + \mathcal{O}\Big(\frac{\beta}{N\alpha} + \frac{\beta \sigma_2^{L/4} 2^T}{M}\Big). \tag{10}$$

*Furthermore, after $T'$ iterations of local averaging, the local models of all agents $m = 1, ..., M$ has the following consensus error (see eq. (34) for its full expression) :*

$$\mathbb{E}\big[\|\theta_{T+T'}^{(m)} - \overline{\theta}_T\|^2\big] \leq \sigma_2^{2T'} \mathcal{O}\Big(1 + \frac{M^4 \beta \alpha}{(1-\sigma_2)^2} + \frac{M \beta \alpha \sigma_2^{L/4} 2^T}{1-\sigma_2}\Big). \tag{11}$$

*Consequently, by choosing $\alpha = \mathcal{O}(\frac{1-\sigma_2}{\sqrt{M}})$, $\beta = \mathcal{O}(1)$, $T = \mathcal{O}(\frac{\sqrt{M} \ln \epsilon^{-1}}{1-\sigma_2})$, $N = \mathcal{O}(\frac{\sqrt{M}}{\epsilon(1-\sigma_2)})$, $L = \mathcal{O}(\frac{\sqrt{M} \ln \epsilon^{-1}}{(1-\sigma_2)^2} + \frac{M}{1-\sigma_2})$, $T' = \mathcal{O}(\frac{1}{1-\sigma_2} \ln \frac{M}{\epsilon(1-\sigma_2)})$ (See the end of Appendix B for the full expressions of these hyperparameters) , we obtain that $\mathbb{E}(\|\theta_{T+T'}^{(m)} - \theta^*\|^2) \leq \epsilon$ for all $m$. The overall communication complexities for synchronizing $\theta_t^{(m)}$ and imporance sampling ratio $\rho$ are respectively $T + T' = \mathcal{O}(\frac{\sqrt{M} \ln \epsilon^{-1}}{1-\sigma_2})$ and $TL = \mathcal{O}((\frac{\sqrt{M} \ln \epsilon^{-1}}{(1-\sigma_2)^3} + \frac{M}{(1-\sigma_2)^2}) \ln \frac{M}{\epsilon(1-\sigma_2)})$. The total sample complexity is $NT = \mathcal{O}(\frac{M \ln \epsilon^{-1}}{\epsilon(1-\sigma_2)^2})$.*

The above theorem shows that our decentralized TDC achieves the sample complexity $\mathcal{O}(\frac{M \ln \epsilon^{-1}}{\epsilon(1-\sigma_2)^2})$, which, in the centralized setting ($M = 1$, $\sigma_2 = 0$), matches $\mathcal{O}(\epsilon^{-1} \ln \epsilon^{-1})$ of centralized TDC for Markovian samples Xu et al. (2019); Xu & Liang (2020) and matches the theoretical lower bound $\mathcal{O}(\epsilon^{-1})$ given in Kaledin et al. (2020) up to a logarithm factor. In addition, the sample complexity is proportional to $M$, which matches the theoretical lower bound of decentralized strongly convex optimization in Scaman et al. (2017). Importantly, the communication complexity $\mathcal{O}(\frac{\sqrt{M} \ln \epsilon^{-1}}{1-\sigma_2})$ for synchronizing $\theta_t^{(m)}$ is substantially lower than the communication complexity $\mathcal{O}((\epsilon^{-1} + \frac{\sqrt{M}}{\sqrt{\epsilon}(1-\sigma_2)}) \ln \epsilon^{-1})$ of decentralized TD(0) Sun et al. (2020) [2] Intuitively, this is because our algorithm adopts mini-batch sampling that significantly reduces the communication frequency, since

---

[2]Since on-policy evaluation does not involve importance sampling ratio $\rho$, we only compare the communication complexity for synchronizing $\theta_t^{(m)}$ which is involved in both on-policy and off-policy evaluation.

communication occurs after collecting a mini-batch of samples to compute the mini-batch updates. Moreover, the communication complexity has a logarithm dependence $\ln \epsilon^{-1}$ on the target accuracy, and this matches the theoretical lower bound of decentralized strongly convex optimization in Scaman et al. (2017). Note that both communication complexity and sample complexity decrease with smaller $\sigma_2 > 0$. Hence, the communication matrix $U$ can be selected with minimum possible $\sigma_2$, under network topology constraint in practice. In addition, our stepsizes $\alpha = \mathcal{O}\left(\frac{1-\sigma_2}{\sqrt{M}}\right)$ and $\beta = \mathcal{O}(1)$ do not scale with $\epsilon$. Although $\alpha$ can be small when $\sigma_2 \approx 1$ and $M$ is large, it is much larger than $\alpha = \min\left[\mathcal{O}(\epsilon), \mathcal{O}\left(\sqrt{\frac{\epsilon}{M}}(1-\sigma_2)\right)\right]$ in decentralized TD (0) Sun et al. (2020) with small $\epsilon$. [3]

Taking a deeper look, Theorem 1 shows that the average model $\overline{\theta}_T$ converges to a small neighborhood of the optimal solution $\theta^*$ at a fast linear convergence rate (10) that matches the convergence rate of centralized TDC Xu et al. (2019); Xu & Liang (2020). In particular, the convergence error is in the order of $\mathcal{O}\left(\frac{\beta}{N\alpha} + \frac{\beta\sigma_2^{L/4}2^T}{M}\right)$, which can be driven arbitrarily close to zero by choosing a sufficiently large mini-batch size $N$ and communication rounds $L$ (with $T$ fixed), and choosing constant-level learning rates $\alpha$, $\beta$. Moreover, the $T'$ steps of extra local model averaging further help all the agents achieve a small consensus error at a linear convergence rate (11). Eqs. (10) and (11) together ensure the fast convergence of all the local model parameters. We want to point out that the $T'$ local averaging steps are critical for establishing fast convergence of local model parameters. Specifically, without the $T'$ local averaging steps, the consensus error $\mathbb{E}\left[\|\theta_T^{(m)} - \overline{\theta}_T\|^2\right]$ would be in the order of at least $\mathcal{O}(1)$, which is constant-level and hence cannot guarantee the local model parameters converge arbitrarily close to the true solution.

**Proof sketch of Theorem 1.** The proof of the theorem is a nontrivial generalization of the analysis of centralized off-policy TDC to the decentralized case. Below, we sketch the technical proof and elaborate on the technical novelties.

- **Step 1.** We first consider an ideal case where the agents can access the exact global importance sampling ratio $\rho_t$ at iteration $t$. In this ideal case, every agent $m$ can replace the estimated global importance sampling ratio $\widehat{\rho}_i^{(m)}$ involved in $A_i^{(m)}$, $B_i^{(m)}$, $\widehat{b}_i^{(m)}$ in the update rules (7) and (8) by the exact value $\rho_i$ (so $A_i^{(m)}$, $B_i^{(m)}$, $\widehat{b}_i^{(m)}$ become $A_i$, $B_i$, $b_i^{(m)}$ respectively) . Then, with the notations defined in Table 1 in Appendix A, the averaged update rules (14) and (15) become

$$\widetilde{\theta}_{t+1} = \overline{\theta}_t + \alpha\left(\widetilde{A}_t\overline{\theta}_t + \widetilde{\overline{b}}_t + \widetilde{B}_t\overline{w}_t\right), \tag{12}$$

$$\widetilde{w}_{t+1} = \overline{w}_t + \beta\left(\widetilde{A}_t\overline{\theta}_t + \widetilde{\overline{b}}_t + \widetilde{C}_t\overline{w}_t\right), \tag{13}$$

  This can be seen as one step of centralized TDC. Hence, we can bound its optimization error terms $\mathbb{E}\left[\|\widetilde{w}_{t+1} - w_t^*\|^2\right]$ and $\mathbb{E}\left[\|\widetilde{\theta}_{t+1} - \theta^*\|^2\right]$.

- **Step 2.** We return to Algorithm 1 and bound its optimization error terms $\mathbb{E}\left[\|\overline{w}_{t+1} - w_{t+1}^*\|^2\right]$ (by bounding $\mathbb{E}\left[\|\overline{w}_{t+1} - w_t^*\|^2\right]$ first) and $\mathbb{E}\left[\|\overline{\theta}_{t+1} - \theta^*\|^2\right]$. This is done by bounding the gap between the centralized updates (12) and (13) (with exact $\rho_t$) and the decentralized updates (14) and (15) (with inexact $\rho_t$). The key is to establish Corollary 2, which strategically controls the gap between the inexact global importance sampling ratio $\widehat{\rho}_t^{(m)}$ and the exact value $\rho_t$. Such a challenge in bounding the importance sampling ratio gap is not seen in the analysis of decentralized TD(0) and centralized TDC.

  To elaborate, note that the locally-averaged importance sampling ratios $\widetilde{\rho}_{t,\ell}^{(m)}$ in eq. (5) exponentially converges to the value $\ln \rho_t$. However, the initial gap $|\ln \rho_t^{(m)} - \ln \rho_t|$ can be numerically large since $\rho_t^{(m)}$ may be a numerically small positive number. To avoid such divergence issue, our proof discusses two complementary cases. Case 1: the quantity $\rho_{\min} := \min_{m \in \mathcal{M}} \rho_t^{(m)} \in [\sigma_2^{L/2}, \rho_{\max}]$. In this case, the proof is straightforward as the initial gap is bounded. Case 2: the quantity $\rho_{\min} \in (0, \sigma_2^{L/2}]$. In this case, we show that the locally-averaged logarithm-ratio $\widetilde{\rho}_{t,L}^{(m)}$ is below a large negative number $\mathcal{O}\left(\frac{\ln \rho_{\min}}{M}\right) \ll 0$ (See eq. (65)). which implies that both the global importance sampling ratio $\rho_t$ and its estimation $\widehat{\rho}_t^{(m)}$ are close

---

[3] $\alpha = \min\left[\mathcal{O}(\epsilon), \mathcal{O}\left(\sqrt{\frac{\epsilon}{M}}(1-\sigma_2)\right)\right]$ is obtained by letting the convergence rate in proposition 2 from Sun et al. (2020) to be smaller than $\epsilon$.

to zero. In both cases, $\{\widehat{\rho}_t^{(m)}\}_{m=1}^M$ converge exponentially fast to $\rho_t$ as $L$ increases. This prove eq. (10). To the best of our knowledge, this technique for bounding the estimation error of the global importance sampling ratio has not been developed in the existing literature.

- **Step 3.** Finally, we prove the consensus error (11). Although the consensus error exponentially decays during the $T'$ extra local average steps in Algorithm 1, it is non-trivial to bound the initial consensus error $\|\Delta\Theta_T\|_F$ of the $T'$ local average iterations (see eq. (34)), which is caused by the $T$ decentralized TDC steps. To bound this error, note that each decentralized TDC step consists of both local averaging and TDC update, which makes the consensus error $\|\Delta\Theta_t\|_F$ diminishes geometrically fast with a noise term $\sum_{m=1}^M \|h_m\|$ (see eq. (30)). Such a noise term is induced by the TDC update and hence its bound depends on both the consensus error and the model estimation error in eq. (10). We need to apply these correlated bounds iteratively for $T$ iterations to bound the initial consensus error $\|\Delta\Theta_T\|_F$.

## 5 Experiments

### 5.1 Simulated Multi-Agent Networks

We simulate a multi-agent MDP with 10 decentralized agents. The shared state space contains 10 states and each agent can take 2 actions. All behavior policies are uniform policies (i.e., each agent takes all actions with equal probability), and the target policies are obtained by first perturbing the corresponding behavior policies with Gaussian noises sampled from $\mathcal{N}(0, 0.05)$ and then performing a proper normalization. The entries of the transition kernel and the reward functions are independently generated from the uniform distribution on $[0, 1]$ (with proper normalization for the transition kernel). We generate all state features with dimension 5 independently from the standard Gaussian distribution and normalize them to have unit norm. The discount factor is $\gamma = 0.95$.

We consider two types of network topologies: a fully connected network with communication matrix $U$ having diagonal entries 0.8 and off-diagonal entries $1/45$, and a ring network with communication matrix $U$ having diagonal entries 0.8 and entries 0.1 for adjacent agents. We implement and compare two algorithms in these networks: the decentralized TD(0) with batch size $N = 1$ (used in Sun et al. (2020)) and our decentralized TDC with batch sizes $N = 1, 10, 20, 50, 100$. Note that the original decentralized TD(0) is simply eq. (7) with setting $w_t^{(m)} \equiv 0$ and $\rho_i \equiv 1$ in the definition of $A_i^{(m)}$, $B_i^{(m)}$ and $\widehat{b}_i^{(m)}$, which only works for on-policy evaluation. To adapt decentralized TD(0) to off-policy evaluation, we simply use $\widehat{\rho}_i^{(m)}$ computed by eqs. (5) and (6) with $L = 3$.

**Effect of Batch size:** We test these algorithms with varying batch size $N$ and compare their sample and communication complexities. We set learning rate $\alpha = 0.2$ for the decentralized TD(0) and $\alpha = 0.2 * N$, $\beta = 0.002 * N$ for our decentralized TDC with varying batch sizes $N = 1, 10, 20, 50, 100$. Both algorithms use $L = 3$ communication rounds for synchronizing $\widehat{\rho}_t^{(m)}$. All algorithms are repeated 100 times using a fixed set of 100 MDP trajectories, each of which has 20k Markovian samples.

We first implement these algorithms in the fully connected network. Figure 1 plots the relative convergence error $\|\bar{\theta}_t - \theta^*\|/\|\theta^*\|$ v.s. the number of samples ($tN$) and the number of communication rounds ($t$) . For each curve, its upper and lower envelopes denote the 95% and 5% percentiles of the 100 convergence errors, respectively. It can be seen that our decentralized TDC with different batch sizes achieve comparable sample complexity to that of the decentralized TD(0), demonstrating the sample-efficiency of our algorithms. On the other hand, our decentralized TDC requires much less communication complexities than the decentralized TD(0) with $N \geq 10$ , and the required communication becomes lighter as batch size increases. All these results match our theoretical analysis well.

We further implement these algorithms in the ring network. The comparison results are exactly the same as those in Figure 1, since the update rule of $\bar{\theta}_t$ does not rely on the network topology under exact global importance sampling.

**Effect of Communication Rounds:** We test our decentralized TDC using varying communication rounds $L = 1, 3, 5, 7$. We use a fixed batch size $N = 100$ and set learning rates $\alpha = 5$, $\beta = 0.05$, and repeat each

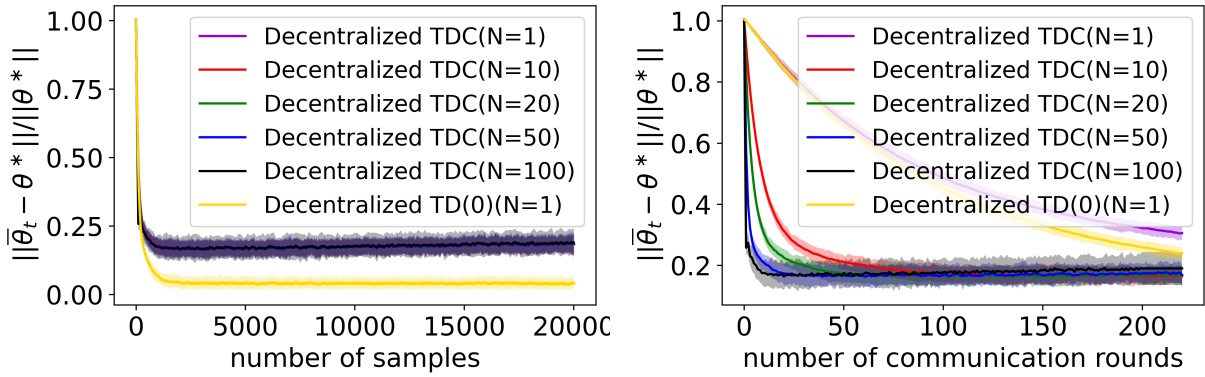

Figure 1: Comparison between decentralized TDC with varying batch sizes and decentralized TD(0).

algorithm 100 times using the set of 100 MDP trajectories. We also implement the decentralized TDC with exact global importance sampling ratios as a baseline. Figure 2 plots the relative convergence error v.s. the number of communication rounds ($t$) in the fully-connected network (Left) and ring network (Right). It can be seen that in both networks, the asymptotic convergence error of the decentralized TDC with inexact $\rho$ decreases as the number of communication rounds $L$ for synchronizing the global importance sampling ratio increases. In particular, with $L = 1$, decentralized TDC diverges asymptotically due to inaccurate estimation of the global importance sampling ratio. As $L$ increases to more than 5, the convergence error is as small as that under exact global importance sampling.

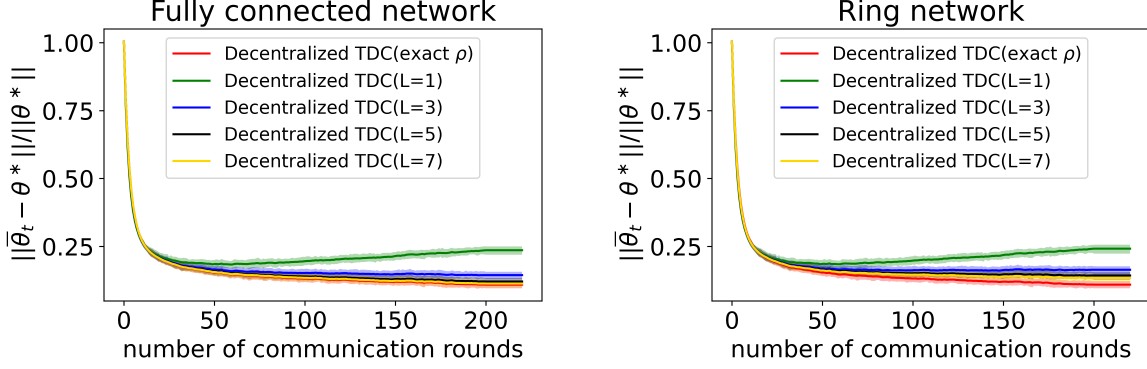

Figure 2: Effect of communication rounds $L$ on asymptotic convergence error.

We further plot the maximum relative consensus error among all agents $\max_m \|\theta_t^{(m)} - \overline{\theta}_t\| / \|\overline{\theta}^*\|$ v.s. the number of communication rounds ($t$) in the fully-connected network (Left) and ring network (Right) in Figure 3, where the tails in both figures correspond to the extra $T' = 20$ local model averaging steps. In both networks, one can see that the consensus error decreases as $L$ increases, and the extra local model averaging steps are necessary to achieve consensus. Moreover, it can be seen that the consensus errors achieved in the fully connected network are slightly smaller than those achieved in the ring network, as denser connections facilitate achieving the global consensus.

## 5.2 Two-Agent Cliff Navigation Problem

In this subsection, we test our algorithms in solving a two-agent Cliff Navigation problem Qiu et al. (2021) in a grid-world environment. This problem is adapted from its single-agent version (see Example 6.6 of Sutton

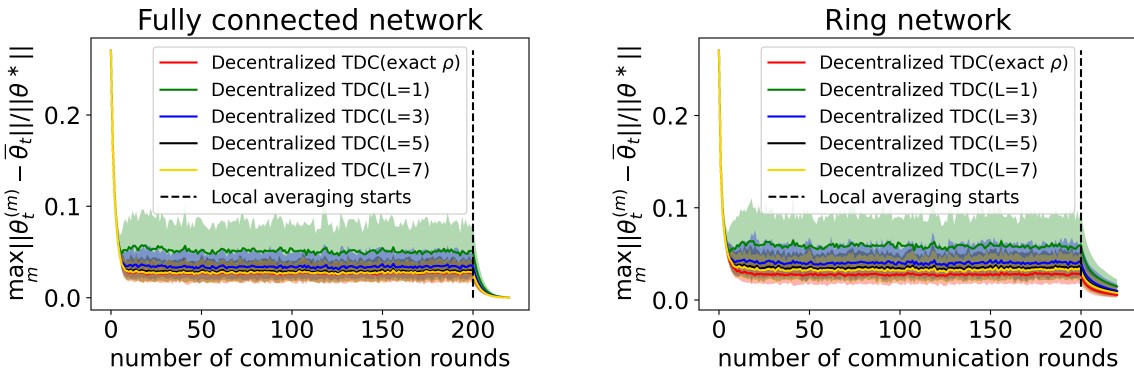

Figure 3: Effect of communication rounds $L$ on consensus error.

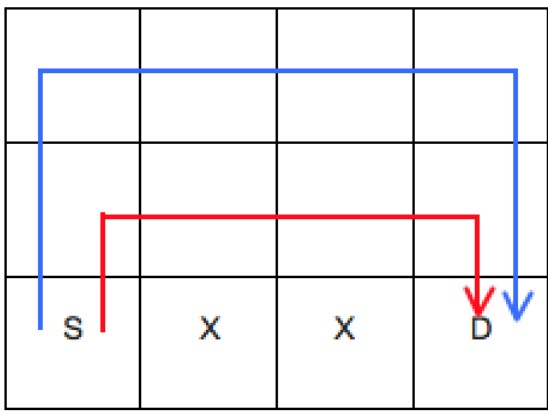

Figure 4: Two-agent cliff navigation. ("S", "X", "D" denote starting point, cliff and destination respectively. The optimal path is shown in red.)

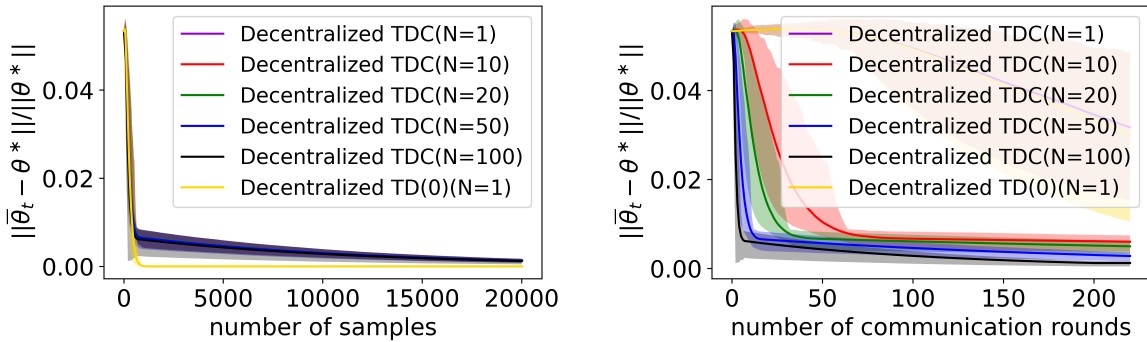

Figure 5: Results on two-agent cliff navigation problem.

& Barto (2018)). As illustrated in Figure 4, two agents start from the starting point "S" on a $3 \times 4$ grid and aim to reach the destination "D". Here, global state is defined as the joint location of the two agents, and there are in total $(3 \times 4)^2 = 144$ global states. In most states, an agent can choose to move up, down, left or right by one step and receives $-1$ reward. However, once an agent falls into the cliff "X", it will return to the starting point "S" and receive $-100$ reward. When an agent reaches "D", it will always stay at "D", and

receives 0 reward if the other agent also reaches/stays at "D", or receives $-0.5$ reward otherwise. If an agent is not at "X" or "D" and selects a direction that points outside the grid, then it stays in the previous location and receives $-1$ reward.

We apply the aforementioned algorithms with different batchsizes $N$ to solve this problem. The hyperparameters and the ways to generate behavior policy and target policy are the same as the previous simulation experiment, except that the communication matrix $U$ has diagonal entries 0.7 and off-diagonals 0.3. All algorithms are repeated 100 times using a fixed set of 100 MDP trajectories, each of which has 20k Markovian samples. Figure 5 plots the relative convergence error $\|\bar{\theta}_t - \theta^*\|/\|\theta^*\|$ v.s. the number of samples $(tN)$ and the number of communication rounds $(t)$ . We can see that compared with the decentralized TD(0), our decentralized TDC achieve comparable sample complexities with different batch sizes and much lower communication complexities with $N \geq 10$ . Moreover, the required communication becomes lighter as batch size increases. These properties are similar to those shown in the simulation and thus have generality.

| X | X | X |   |   |   |   | D1 | D2 | D3 |
|---|---|---|---|---|---|---|----|----|----|
|   |   |   |   |   |   |   |    |    | X  |
|   | X |   |   | X |   |   |    |    | X  |
|   |   |   | X |   |   |   |    |    |    |
|   |   |   |   |   |   | X | X  |    |    |
|   |   |   |   | X |   |   | X  |    |    |
|   | X |   |   |   |   |   |    |    |    |
|   |   |   |   |   |   |   | X  |    |    |
|   |   |   |   |   | X |   | X  |    |    |
| S1 | S2 | S3 | X |   |   |   | X  |    |    |

Figure 6: The map for path finding problem. ("S1", "S2", "S3" denote the starting points of the 3 agents. "D1", "D2", "D3" denote their destinations. "X" denotes an obstacle.)

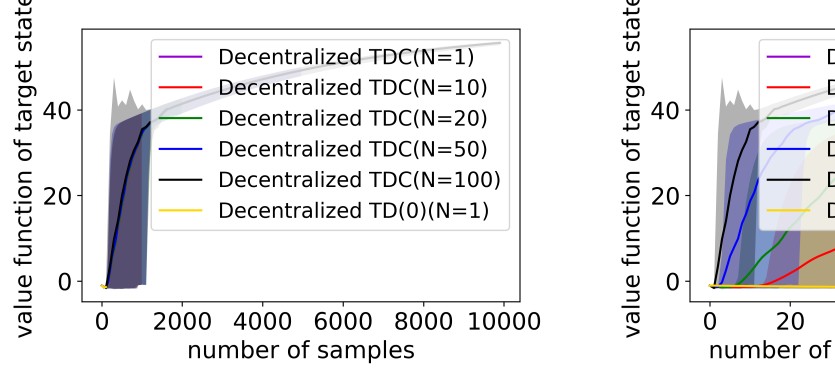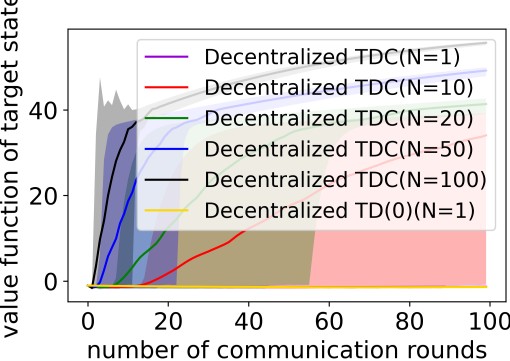

Figure 7: Results on path finding problem.

## 5.3 Application to Path Finding Problem

In this subsection, we test our algorithms in solving a multi-agent path finding problem Ma et al. (2021), which has broad real-world applications including aircraft-towing vehicles Morris et al. (2016), ware-house and office robots Wurman et al. (2008); Veloso et al. (2015), and video games Silver (2005); Ma et al. (2017). As illustrated in Figure 6, three agents start fron the points "S1", "S2", "S3" on a $10 \times 10$ grid and aim to reach the destination "D1", "D2", "D3", respectively. In most cases, an agent can choose to move up, down, left or right by one step or stay and receives -0.075 reward. However, when an agent reaches its destination, it always stay and receives 3 reward if all the agents reach their destinations, or receives 0 reward otherwise. If an agent has not reached its destination and collides with an obstacle "X" or another agent, it receives -0.5 reward.

We apply the aforementioned algorithms with different batchsizes $N$ to solve this problem. The implementation details are mostly the same as those of the previous two-agent cliff navigation problem, with the following differences. (1) The behavior policy selects from all available actions uniformly at random, and the target policy selects from available actions that avoid obstacles uniformly at random; (2) The communication matrix $U$ has diagonal entries 0.6 and off-diagonals 0.2; (3) We use $\alpha = 0.08 * N$ obtained by tuning while $\beta = 0.002 * N$ is the same as aforementioned. (4) The feature vector of any state $s$ (all the agents' locations) is $\phi(s) = [\phi^{(1)}(s), \phi^{(2)}(s), \phi^{(3)}(s)]$ where $\phi^{(m)} \in \{0,1\}^5$ is defined as follows: $\phi_1^{(m)}(s) = 1$, $\phi_2^{(m)}(s) = 1$ or $\phi_3^{(m)}(s) = 1$ if and only if the $m$-th agent reaches its goal, is 1 step away from its goal, or is 1 horizontal step and 1 vertical step away from its goal, respectively. $\phi_4^{(m)}(s) = 1$ if and only if the $m$-th agent has not reached its goal and there is at least one obstacle or other agents in its 8 surrounding grids. $\phi_5^{(m)}(s) = 1$ if and only if the $m$-th agent has not reached its goal and collides with an obstacle "X" or another agent.

Figure 7 plots the value function at the target state where all the agents reach the goal (i.e., the sum of entries $(\theta_t)_1 + (\theta_t)_6 + (\theta_t)_{11}$) v.s. the number of samples ($tN$) and the number of communication rounds ($t$). We can see that all the algorithms converge to the true value $3/(1 - \gamma) = 60$. Compared with the decentralized TD(0), our decentralized TDC achieve comparable sample complexities with different batch sizes and much lower communication complexities with $N \geq 10$. Moreover, the required communication becomes lighter as batch size increases. These properties are similar to those shown in the simulation and thus have generality.

## 6 Conclusion

In this paper, we develop a sample-efficient and communication-efficient decentralized TDC algorithm for multi-agent off-policy evaluation. Our algorithm synchronizes the local importance sampling ratios among the agents and adopts mini-batch stochastic updates to save communication. In particular, it avoids sharing agents' sensitive local information. We prove that the proposed decentralized TDC algorithms achieve a near-optimal sample complexity as well as an optimal communication complexity that improves over the existing decentralized TD(0). In the future, we expect that our algorithm can serve as a fundamental component in the design of advanced policy optimization algorithms for MARL.

### Acknowledgments

The work of Z. Chen and Y. Zhou was supported in part by U.S. National Science Foundation under the Grants CCF-2106216 and DMS-2134223.

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

## A  Notations and Filtration

### A.1  Notations to rewrite update rules in Algorithm 1

We introduce the notations in Table 1 that will be used throughout our proof .

Table 1: List of notations

| Notations | Explanation |
|---|---|
| $\rho_i$, $A_i$, $B_i$, $C_i$ (defined in eq. (3)), $b_i^{(m)} = \rho_i R_i^{(m)} \phi(s_i)$ | These quantities of the $i$-th sample use exact global importance sampling ratio $\rho_i$. |
| $A_i^{(m)} := \widehat{\rho}_i^{(m)} \phi(s_i)(\gamma\phi(s_{i+1}) - \phi(s_i))^\top$ $B_i^{(m)} := \widehat{\rho}_i^{(m)} \phi(s_{i+1})\phi(s_i))^\top$ | Replace $\rho_i$ and $R_i$ in $A_i$, $B_i$ (see eq. (3)) with $\widehat{\rho}_i^{(m)}$ and $R_i^{(m)}$ respectively. |
| $\widehat{b}_i^{(m)} := \widehat{\rho}_i^{(m)} R_i^{(m)} \phi(s_i)$ | Replace $\rho_i$ in $b_i^{(m)}$ with $\widehat{\rho}_i^{(m)}$ (defined by eqs. (5) and (6)). |
| $\overline{\theta}_t = \frac{1}{M} \sum_{m=1}^{M} \theta_t^{(m)}$ | Agents' average model parameter. $\overline{w}_t$ and $\overline{b}_i$ are defined similarly. |
| $\widetilde{A}_t = \frac{1}{N} \sum_{i=tN}^{(t+1)N-1} A_i$ | Minibatch average of global quantity. $\widetilde{B}_t$ and $\widetilde{C}_t$ are defined similarly. |
| $\widetilde{A}_t^{(m)} = \frac{1}{N} \sum_{i=tN}^{(t+1)N-1} A_i^{(m)}$ | Average local quantity over the minibatch at the $t$-th iteration. $\widetilde{B}_t^{(m)}$, $\widetilde{b}_t^{(m)}$ and $\widetilde{\widehat{b}}_t^{(m)}$ are defined similarly. |
| $\overline{\widetilde{b}}_t = \frac{1}{N} \sum_{i=tN}^{(t+1)N-1} \overline{b}_i = \frac{1}{M} \sum_{m=1}^{M} \widetilde{b}_t^{(m)}$ | Average over both agents and minibatch. |
| $A := \mathbb{E}_{\pi_b}[A_i]$, $B := \mathbb{E}_{\pi_b}[B_i]$ $C := \mathbb{E}_{\pi_b}[C_i]$, $\overline{b} := \mathbb{E}_{\pi_b}[\overline{b}_i]$ | Expected quantities. |
| $\theta^* = -A^{-1}b$ | The optimal model parameter |
| $w_t^* = -C^{-1}(A\overline{\theta}_t + b)$ | The optimal auxiliary parameter corresponding to $\overline{\theta}_t$. |
| $\|A\|_F := \left(\sum_{i,j} A_{i,j}^2\right)^{1/2}$ | Frobenius norm. |

Then, by averaging the update rules (7) and (8) over $m$, we obtain the following update rules of the model average $\overline{\theta}_t, \overline{w}_t$.

$$\overline{\theta}_{t+1} = \overline{\theta}_t + \frac{\alpha}{M} \sum_{m=1}^{M} \left(\widetilde{A}_t^{(m)} \theta_t^{(m)} + \widetilde{\widehat{b}}_t^{(m)} + \widetilde{B}_t^{(m)} w_t^{(m)}\right), \tag{14}$$

$$\overline{w}_{t+1} = \overline{w}_t + \frac{\beta}{M} \sum_{m=1}^{M} \left(\widetilde{A}_t^{(m)} \theta_t^{(m)} + \widetilde{\widehat{b}}_t^{(m)} + \widetilde{C}_t w_t^{(m)}\right). \tag{15}$$

### A.2 Filtration

Define the filtration $\mathcal{F}_t = \sigma\left(\{s_{t'}, a_{t'}\}_{t'=1}^{tN-1} \cup \{s_{tN}\}\right)$. Then,

$$\widetilde{A}_t, \widetilde{B}_t, \widetilde{C}_t, \widetilde{b}_t^{(m)}, \overline{\widetilde{b}}_t, \widetilde{A}_t^{(m)}, \widetilde{B}_t^{(m)}, \widehat{\widetilde{b}}_t^{(m)} \in \mathcal{F}_{t+1}/\mathcal{F}_t, \qquad \theta_t^{(m)}, \overline{\theta}_t, w_t^{(m)}, \overline{w}_t, w_t^* \in \mathcal{F}_t/\mathcal{F}_{t-1}.$$

# B    Proof of Theorem 1

**Step 1: Bounding optimization error for ideal case.** We first consider an ideal case where the agents can access the exact global importance sampling ratio $\rho_t$ at iteration $t$. In this ideal case, every agent $m$ can replace the estimated global importance sampling ratio $\widehat{\rho}_i^{(m)}$ involved in $A_i^{(m)}, B_i^{(m)}, \widehat{b}_i^{(m)}$ in the update rules (7) and (8) by the exact value $\rho_i$. Then, with the notations defined in Appendix A, the averaged update rules (14) and (15) respectively become eqs. (12) and (13) as repeated below

$$\widetilde{\theta}_{t+1} = \overline{\theta}_t + \alpha\left(\widetilde{A}_t\overline{\theta}_t + \overline{\widetilde{b}}_t + \widetilde{B}_t\overline{w}_t\right),$$

$$\widetilde{w}_{t+1} = \overline{w}_t + \beta\left(\widetilde{A}_t\overline{\theta}_t + \overline{\widetilde{b}}_t + \widetilde{C}_t\overline{w}_t\right).$$

The aim of Step 1 is to bound the following optimization errors of $\widetilde{w}_{t+1}$, $\widetilde{\theta}_{t+1}$ obtained by the centralized update rules (13) and (12) respectively.

$$\mathbb{E}\left[\|\widetilde{w}_{t+1} - w_t^*\|^2 \big| \mathcal{F}_t\right]$$
$$= \|\overline{w}_t - w_t^*\|^2 + 2\beta \underbrace{(\overline{w}_t - w_t^*)^\top \mathbb{E}\left[\widetilde{A}_t\overline{\theta}_t + \overline{\widetilde{b}}_t + \widetilde{C}_t\overline{w}_t \big| \mathcal{F}_t\right]}_{(I)} + \beta^2 \underbrace{\mathbb{E}\left[\left\|\widetilde{A}_t\overline{\theta}_t + \overline{\widetilde{b}}_t + \widetilde{C}_t\overline{w}_t\right\|^2 \big| \mathcal{F}_t\right]}_{(II)}, \qquad (16)$$

$$\mathbb{E}\left[\|\widetilde{\theta}_{t+1} - \theta^*\|^2 \big| \mathcal{F}_t\right]$$
$$= \|\overline{\theta}_t - \theta^*\|^2 + 2\alpha \underbrace{(\overline{\theta}_t - \theta^*)^\top \mathbb{E}\left[\widetilde{A}_t\overline{\theta}_t + \overline{\widetilde{b}}_t + \widetilde{B}_t\overline{w}_t \big| \mathcal{F}_t\right]}_{(III)} + \alpha^2 \underbrace{\mathbb{E}\left[\left\|\widetilde{A}_t\overline{\theta}_t + \overline{\widetilde{b}}_t + \widetilde{B}_t\overline{w}_t\right\|^2 \big| \mathcal{F}_t\right]}_{(IV)}. \qquad (17)$$

The above four terms (I)-(IV) are respectively bounded below.

$$(I) = (\overline{w}_t - w_t^*)^\top \mathbb{E}\left[\widetilde{A}_t\overline{\theta}_t + \overline{\widetilde{b}}_t + \widetilde{C}_t\overline{w}_t \big| \mathcal{F}_t\right]$$
$$\overset{(i)}{=} 2(\overline{w}_t - w_t^*)^\top \mathbb{E}\left[\widetilde{C}_t - C \big| \mathcal{F}_t\right](\overline{w}_t - w_t^*) + 2(\overline{w}_t - w_t^*)^\top C(\overline{w}_t - w_t^*) + 2(\overline{w}_t - w_t^*)^\top$$
$$\mathbb{E}\left[(\widetilde{A}_t - \widetilde{C}_t C^{-1}A)\overline{\theta}_t + \overline{\widetilde{b}}_t - \widetilde{C}_t C^{-1}b \big| \mathcal{F}_t\right]$$
$$\overset{(ii)}{\le} 2\left\|\mathbb{E}\left[\widetilde{C}_t - C \big| \mathcal{F}_t\right]\right\|_F \|\overline{w}_t - w_t^*\|^2 - 2\lambda_2\|\overline{w}_t - w_t^*\|^2 + \lambda_2\|\overline{w}_t - w_t^*\|^2 + \frac{3}{\lambda_2}\mathbb{E}\left[\left\|\overline{\widetilde{b}}_t - \widetilde{C}_t C^{-1}b\right\|^2 \big| \mathcal{F}_t\right]$$
$$+ \frac{3}{\lambda_2}\mathbb{E}\left[\left\|(\widetilde{A}_t - \widetilde{C}_t C^{-1}A)\right\|_F^2 \big| \mathcal{F}_t\right]\|\overline{\theta}_t - \theta^*\|^2 + \frac{3}{\lambda_2}\mathbb{E}\left[\left\|\widetilde{A}_t - \widetilde{C}_t C^{-1}A\right\|_F^2 \big| \mathcal{F}_t\right]\|\theta^*\|^2$$
$$\overset{(iii)}{\le} \left(\frac{4\nu\rho_{\max}}{N(1-\delta)} - \lambda_2\right)\|\overline{w}_t - w_t^*\|^2 + \frac{96\rho_{\max}^2(\nu+1)}{N\lambda_2(1-\delta)}\left(1 + \frac{1}{\lambda_2}\right)^2\left(\|\overline{\theta}_t - \theta^*\|^2 + \|\theta^*\|^2 + R_{\max}^2\right) \qquad (18)$$

where (i) uses the notation that $w_t^* = -C^{-1}(A\overline{\theta}_t + b)$, (ii) uses $\lambda_2 = -\lambda_{\max}(C)$, the inequality that $\|Ax\| \le \|A\|_F\|x\|$ for any matrix $A$ and vector $x$ , and the inequality that $2a_1^\top a_2 \le \sigma^{-1}\|a_1\|^2 + \sigma\|a_2\|^2$ for any $a_1, a_2 \in \mathbb{R}^d$ and $\sigma > 0$, and applies Jensen's inequality to convex functions $\|\cdot\|$ and $\|\cdot\|^2$ and uses eq. (44), (iv) uses eqs. (47) and (48).

$$(II) = \mathbb{E}\left[\left\|\widetilde{A}_t\overline{\theta}_t + \overline{\widetilde{b}}_t + \widetilde{C}_t\overline{w}_t\right\|^2 \big| \mathcal{F}_t\right]$$

$$\overset{(i)}{=} \mathbb{E}\big[\big\|(\widetilde{A}_t - \widetilde{C}_t C^{-1}A)(\overline{\theta}_t - \theta^*) + (\widetilde{A}_t - \widetilde{C}_t C^{-1}A)\theta^* + \overline{\overline{b}}_t + \widetilde{C}_t(\overline{w}_t - w_t^*) - \widetilde{C}_t C^{-1}b\big\|^2\big|\mathcal{F}_t\big]$$

$$\overset{(ii)}{\leq} 4\mathbb{E}\big[\|\widetilde{A}_t - \widetilde{C}_t C^{-1}A\|_F^2\big|\mathcal{F}_t\big]\|\overline{\theta}_t - \theta^*\|^2 + 4\|\overline{w}_t - w_t^*\|^2$$

$$\qquad + 4\mathbb{E}\big[\|\widetilde{A}_t - \widetilde{C}_t C^{-1}A\|_F^2\big|\mathcal{F}_t\big]\|\theta^*\|^2 + 4\mathbb{E}\big[\|\overline{\overline{b}}_t - \widetilde{C}_t C^{-1}b\|^2\big|\mathcal{F}_t\big]$$

$$\overset{(iii)}{\leq} \frac{128\rho_{\max}^2(\nu+1)}{N(1-\delta)}\Big(1+\frac{1}{\lambda_2}\Big)^2\big(\|\overline{\theta}_t - \theta^*\|^2 + \|\theta^*\|^2 + R_{\max}^2\big) + 4\|\overline{w}_t - w_t^*\|^2 \qquad (19)$$

where (i) uses the notation that $w_t^* = -C^{-1}(A\overline{\theta}_t + b)$, (ii) uses $\|\sum_{k=1}^4 a_k\|^2 \leq 4\sum_{k=1}^4 \|a_k\|^2$ for any $a_1, a_2, a_3, a_4 \in \mathbb{R}^d$ and eq. (41), and $\|Ax\| \leq \|A\|_F\|x\|$ for any matrix $A$ and vector $x$, (iii) uses eqs. (47) and (48).

$$(III) = (\overline{\theta}_t - \theta^*)^\top \mathbb{E}\big[\widetilde{A}_t\overline{\theta}_t + \overline{\overline{b}}_t + \widetilde{B}_t\overline{w}_t\big|\mathcal{F}_t\big]$$

$$\overset{(i)}{=} 2(\overline{\theta}_t - \theta^*)^\top \mathbb{E}\big[(\widetilde{A}_t - \widetilde{B}_t C^{-1}A - A^\top C^{-1}A)(\overline{\theta}_t - \theta^*) + A^\top C^{-1}A(\overline{\theta}_t - \theta^*)$$

$$\qquad + (\widetilde{A}_t - \widetilde{B}_t C^{-1}A - A^\top C^{-1}A)\theta^* + \widetilde{B}_t(\overline{w}_t - w_t^*) + \overline{\overline{b}}_t - b - (\widetilde{B}_t - B)C^{-1}b\big|\mathcal{F}_t\big]$$

$$\overset{(ii)}{\leq} 2\mathbb{E}\big[\|\widetilde{A}_t - \widetilde{B}_t C^{-1}A - A^\top C^{-1}A\|\big|\mathcal{F}_t\big]\|\overline{\theta}_t - \theta^*\|^2 - 2\lambda_1\|\overline{\theta}_t - \theta^*\|^2 + \lambda_1\|\overline{\theta}_t - \theta^*\|^2 + \|\widetilde{B}_t(\overline{w}_t - w_t^*)\|^2$$

$$\qquad + \frac{4}{\lambda_1}\mathbb{E}\big[\|\widetilde{A}_t - \widetilde{B}_t C^{-1}A - A^\top C^{-1}A\|^2\big|\theta^*\|^2 + \|\overline{\overline{b}}_t - b\|^2 + \|(\widetilde{B}_t - B)C^{-1}b\|^2\big|\mathcal{F}_t\big]$$

$$\overset{(iii)}{\leq} \Big(\frac{64\rho_{\max}^2(\nu+1)}{N(1-\delta)}\Big(1+\frac{\rho_{\max}}{\lambda_2}\Big)^2 - \lambda_1\Big)\|\overline{\theta}_t - \theta^*\|^2 + \frac{4\rho_{\max}^2}{\lambda_1}\|\overline{w}_t - w_t^*\|^2$$

$$\qquad + \frac{32\rho_{\max}^2(\nu+1)}{N\lambda_1(1-\delta)}\Big(4\Big(1+\frac{\rho_{\max}}{\lambda_2}\Big)^2\|\theta^*\|^2 + R_{\max}^2 + \frac{\rho_{\max}R_{\max}}{\lambda_2}\Big) \qquad (20)$$

where (i) uses the notations that $w_t^* = -C^{-1}(A\overline{\theta}_t + b)$ and that $b = -A\theta^*$, and the relation that $C - B = A^\top$, (ii) uses the notation that $\lambda_1 = -\lambda_{\max}(A^\top C^{-1}A)$ and the inequality that $2a_1^\top a_2 \leq \sigma^{-1}\|a_1\|^2 + \sigma\|a_2\|^2$ for any $a_1, a_2 \in \mathbb{R}^d$ and $\sigma > 0$, and applies Jensen's inequality to the convex functions $\|\cdot\|$ and $\|\cdot\|^2$, (iii) uses eqs. (40), (42), (43), (45), (46) and (49).

$$(IV) = \mathbb{E}\big[\|\widetilde{A}_t\overline{\theta}_t + \overline{\overline{b}}_t + \widetilde{B}_t\overline{w}_t\|^2\big|\mathcal{F}_t\big]$$

$$\overset{(i)}{=} \mathbb{E}\big[\|(\widetilde{A}_t - \widetilde{B}_t C^{-1}A)(\overline{\theta}_t - \theta^*) + \overline{\overline{b}}_t - b + \widetilde{B}_t(\overline{w}_t - w_t^*)$$

$$\qquad + (\widetilde{A}_t - \widetilde{B}_t C^{-1}A - A^\top C^{-1}A)\theta^* - (\widetilde{B}_t - B)C^{-1}b\|^2\big|\mathcal{F}_t\big]$$

$$\overset{(ii)}{\leq} 10\mathbb{E}\big[\|\widetilde{A}_t\|_F^2 + \|\widetilde{B}_t C^{-1}A\|_F^2\big|\mathcal{F}_t\big]\|\overline{\theta}_t - \theta^*\|^2 + 5\mathbb{E}\big[\|\widetilde{A}_t - \widetilde{B}_t C^{-1}A - A^\top C^{-1}A\|_F^2\big|\mathcal{F}_t\big]\|\theta^*\|^2$$

$$\qquad + 5\mathbb{E}\big[\|\overline{\overline{b}}_t - b\|^2\big|\mathcal{F}_t\big] + 5\mathbb{E}\big[\|\widetilde{B}_t\|_F^2\big|\mathcal{F}_t\big]\|\overline{w}_t - w_t^*\|^2 + 5\mathbb{E}\big[\|\widetilde{B}_t - B\|_F^2\big|\mathcal{F}_t\big]\|C^{-1}b\|^2$$

$$\overset{(iii)}{\leq} 40\rho_{\max}^2\Big(1+\frac{\rho_{\max}}{\lambda_2}\Big)^2\|\overline{\theta}_t - \theta^*\|^2 + 5\rho_{\max}^2\|\overline{w}_t - w_t^*\|^2$$

$$\qquad + \frac{160\rho_{\max}^2(\nu+1)}{N(1-\delta)}\Big(1+\frac{\rho_{\max}}{\lambda_2}\Big)^2\big(\|\theta^*\|^2 + R_{\max}^2\big) \qquad (21)$$

where (i) uses the notations that $w_t^* = -C^{-1}(A\overline{\theta}_t + b)$ and that $b = -A\theta^*$, and the relation that $C - B = A^\top$, (ii) uses $\|Ax\| \leq \|A\|_F\|x\|$ for any matrix $A$ and vector $x$ and $\|\sum_{k=1}^K a_k\|^2 \leq K\sum_{k=1}^K \|a_k\|^2$ for any $a_k \in \mathbb{R}^d$ and eqs. (40), (45), (46) and (49), (iii) uses eqs. (39), (40), (42) and (43) and $(1 + \rho_{\max}^2/\lambda_2^2) \leq (1 + \rho_{\max}/\lambda_2)^2$. Substituting the above terms (18)-(21) into eqs. (16) and (17) gives the following upper bounds of $\mathbb{E}\big[\|\widetilde{w}_{t+1} - w_t^*\|^2\big|\mathcal{F}_t\big]$ and $\mathbb{E}\big[\|\widetilde{\theta}_{t+1} - \theta^*\|^2\big|\mathcal{F}_t\big]$.

$$\mathbb{E}\big[\|\widetilde{w}_{t+1} - w_t^*\|^2\big|\mathcal{F}_t\big]$$

$$\leq \Big[1 + 2\beta\Big(\frac{4\nu\rho_{\max}}{N(1-\delta)} - \lambda_2\Big) + 4\beta^2\Big]\|\overline{w}_t - w_t^*\|^2$$

$$+ \frac{64\rho_{\max}^2(\nu+1)}{N(1-\delta)}\Big(1+\frac{1}{\lambda_2}\Big)^2\Big(\frac{3\beta}{\lambda_2}+2\beta^2\Big)\big(\|\overline{\theta}_t - \theta^*\|^2 + \|\theta^*\|^2 + R_{\max}^2\big)$$

$$\overset{(i)}{\leq} \Big(1 - \frac{\beta\lambda_2}{2}\Big)\|\overline{w}_t - w_t^*\|^2 + \frac{320\beta\rho_{\max}^2(\nu+1)}{N\lambda_2(1-\delta)}\Big(1+\frac{1}{\lambda_2}\Big)^2\big(\|\overline{\theta}_t - \theta^*\|^2 + \|\theta^*\|^2 + R_{\max}^2\big), \qquad (22)$$

where (i) uses $N \geq \frac{8\nu\rho_{\max}}{\lambda_2(1-\delta)}$ and $\beta \leq \min\big(\frac{\lambda_2}{8}, \frac{1}{\lambda_2}\big)$.

$$\mathbb{E}\big[\|\widetilde{\theta}_{t+1} - \theta^*\|^2\big|\mathcal{F}_t\big]$$

$$\leq \Big(1 + \frac{128\alpha\rho_{\max}^2(\nu+1)}{N(1-\delta)}\Big(1+\frac{\rho_{\max}}{\lambda_2}\Big)^2 - 2\alpha\lambda_1 + 40\alpha^2\rho_{\max}^2\Big(1+\frac{\rho_{\max}}{\lambda_2}\Big)^2\Big)\|\overline{\theta}_t - \theta^*\|^2$$

$$+ \alpha\rho_{\max}^2\Big(\frac{8}{\lambda_1}+5\alpha\Big)\|\overline{w}_t - w_t^*\|^2 + \frac{64\alpha\rho_{\max}^2(\nu+1)}{N\lambda_1(1-\delta)}\Big(4\Big(1+\frac{\rho_{\max}}{\lambda_2}\Big)^2\|\theta^*\|^2 + R_{\max}^2 + \frac{\rho_{\max}R_{\max}}{\lambda_2}\Big)$$

$$+ \frac{160\alpha^2\rho_{\max}^2(\nu+1)}{N(1-\delta)}\Big(1+\frac{\rho_{\max}}{\lambda_2}\Big)^2\big(\|\theta^*\|^2 + R_{\max}^2\big)$$

$$\overset{(i)}{\leq} \Big(1 - \frac{\alpha\lambda_1}{2}\Big)\|\overline{\theta}_t - \theta^*\|^2 + \frac{13\alpha\rho_{\max}^2}{\lambda_1}\|\overline{w}_t - w_t^*\|^2 + \frac{224\alpha\rho_{\max}^2(\nu+1)}{N\lambda_1(1-\delta)}\Big(1+\frac{\rho_{\max}}{\lambda_2}\Big)^2\big(4\|\theta^*\|^2 + 2R_{\max}^2 + 1\big), \tag{23}$$

where (i) uses $N \geq \frac{128\rho_{\max}^2(\nu+1)}{\lambda_1(1-\delta)}\big(1+\frac{\rho_{\max}}{\lambda_2}\big)^2$, $\alpha \leq \min\big[\frac{\lambda_1}{40\rho_{\max}^2}\big(1+\frac{\rho_{\max}}{\lambda_2}\big)^{-2}, \frac{1}{\lambda_1}\big]$ and $\frac{\rho_{\max}R_{\max}}{\lambda_2} \leq \big(1+\frac{\rho_{\max}}{\lambda_2}\big)^2 + R_{\max}^2$.

**Step 2: Bounding optimization error of Algorithm 1.** With the above upper bounds (22) and (23), we derive the upper bounds of $\mathbb{E}\big[\|\overline{w}_{t+1} - w_t^*\|^2\big|\mathcal{F}_t\big]$, $\mathbb{E}\big[\|\overline{w}_{t+1} - w_{t+1}^*\|^2\big|\mathcal{F}_t\big]$ and $\mathbb{E}\big[\|\overline{\theta}_{t+1} - \theta^*\|^2\big|\mathcal{F}_t\big]$ as follows.

$$\mathbb{E}\big[\|\overline{w}_{t+1} - w_t^*\|^2\big|\mathcal{F}_t\big]$$

$$\overset{(i)}{\leq} \Big(1 + \frac{1}{6/(\beta\lambda_2) - 3}\Big)\mathbb{E}\big(\|\widetilde{w}_t - w_t^*\|^2\big|\mathcal{F}_t\big) + \Big(1 + \frac{6}{\beta\lambda_2} - 3\Big)\mathbb{E}\big(\|\overline{w}_t - \widetilde{w}_t\|^2\big|\mathcal{F}_t\big)$$

$$\overset{(ii)}{\leq} \frac{6 - 2\beta\lambda_2}{6 - 3\beta\lambda_2}\Big[\Big(1 - \frac{\beta\lambda_2}{2}\Big)\|\overline{w}_t - w_t^*\|^2 + \frac{320\beta\rho_{\max}^2(\nu+1)}{N\lambda_2(1-\delta)}\Big(1+\frac{1}{\lambda_2}\Big)^2\big(\|\overline{\theta}_t - \theta^*\|^2 + \|\theta^*\|^2 + R_{\max}^2\big)\Big]$$

$$+ \frac{6}{\beta\lambda_2}\mathbb{E}\Big[\Big\|\frac{\beta}{M}\sum_{m=1}^M \big[(\widetilde{A}_t^{(m)} - \widetilde{A}_t)\theta_t^{(m)} + \widetilde{\overline{b}}_t^{(m)} - \widetilde{b}_t^{(m)}\big]\Big\|^2\Big|\mathcal{F}_t\Big]$$

$$\overset{(iii)}{\leq} \Big(1 - \frac{\beta\lambda_2}{3}\Big)\|\overline{w}_t - w_t^*\|^2 + \frac{4}{3}\frac{320\beta\rho_{\max}^2(\nu+1)}{N\lambda_2(1-\delta)}\Big(1+\frac{1}{\lambda_2}\Big)^2\big(\|\overline{\theta}_t - \theta^*\|^2 + \|\theta^*\|^2 + R_{\max}^2\big)$$

$$+ \frac{6\beta}{\lambda_2}\mathbb{E}\Big[\frac{1}{M}\sum_{m=1}^M \big(\|(\widetilde{A}_t^{(m)} - \widetilde{A}_t)\theta_t^{(m)} + \widetilde{\overline{b}}_t^{(m)} - \widetilde{b}_t^{(m)}\|^2\big)\Big|\mathcal{F}_t\Big]$$

$$\overset{(iv)}{\leq} \Big(1 - \frac{\beta\lambda_2}{3}\Big)\|\overline{w}_t - w_t^*\|^2 + \frac{427\beta\rho_{\max}^2(\nu+1)}{N\lambda_2(1-\delta)}\Big(1+\frac{1}{\lambda_2}\Big)^2\big(\|\overline{\theta}_t - \theta^*\|^2 + \|\theta^*\|^2 + R_{\max}^2\big)$$

$$+ \frac{12\beta\sigma_2^{L/4}}{M\lambda_2}\Big(16\max_{m\in\mathcal{M}}(\|\theta_0^{(m)}\| + \|w_0^{(m)}\| + R_{\max})^2[1 + \beta(2\rho_{\max}+3)]^{2t} + R_{\max}^2\Big), \tag{24}$$

where (i) uses the inequality that $\|a_1 + a_2\|^2 \leq (1+\sigma)\|a_1\|^2 + (1+\sigma^{-1})\|a_2\|^2$ for any $a_1, a_2 \in \mathbb{R}^d$ and $\sigma > 0$, (ii) uses eqs. (15), (13) and (22), (iii) applies Jensen's inequality to the convex function $\|\cdot\|^2$ and uses $\beta \leq \frac{1}{\lambda_2}$, (iv) uses the condition that $\beta \leq \frac{1}{\lambda_2}$ which implies that $1 + \frac{1}{6/(\beta\lambda_2)-3} \leq 2$, the inequality that $\|a_1 + a_2\|^2 \leq 2\|a_1\|^2 + 2\|a_2\|^2$ for any $a_1, a_2 \in \mathbb{R}^d$, and eqs. (52), (54) and (68).

$$\mathbb{E}\big[\|\overline{w}_{t+1} - w_{t+1}^*\|^2\big|\mathcal{F}_t\big]$$

$$\overset{(i)}{\leq} \Big(1 + \frac{1}{2[3/(\beta\lambda_2)-1]}\Big)\mathbb{E}\big[\|\overline{w}_{t+1} - w_t^*\|^2\big|\mathcal{F}_t\big] + \big[1 + 2\big(3/(\beta\lambda_2)-1\big)\big]\mathbb{E}\big[\|w_{t+1}^* - w_t^*\|^2\big|\mathcal{F}_t\big]$$

$$\overset{(ii)}{\le} \frac{6/(\beta\lambda_2)-1}{2[3/(\beta\lambda_2)-1]}\Big[\Big(1-\frac{\beta\lambda_2}{3}\Big)\|\overline{w}_t - w_t^*\|^2 + \frac{427\beta\rho_{\max}^2(\nu+1)}{N\lambda_2(1-\delta)}\Big(1+\frac{1}{\lambda_2}\Big)^2(\|\overline{\theta}_t-\theta^*\|^2+\|\theta^*\|^2+R_{\max}^2)$$

$$+ \frac{12\beta\sigma_2^{L/4}}{M\lambda_2}\Big(16\max_{m\in\mathcal{M}}(\|\theta_0^{(m)}\|+\|w_0^{(m)}\|+R_{\max})^2[1+\beta(2\rho_{\max}+3)]^{2t}+R_{\max}^2\Big)\Big]$$

$$+ \frac{6}{\beta\lambda_2}\mathbb{E}\big[\|C^{-1}A(\overline{\theta}_{t+1}-\overline{\theta}_t)\|^2\big|\mathcal{F}_t\big]$$

$$\overset{(iii)}{\le} \Big(1-\frac{\beta\lambda_2}{6}\Big)\|\overline{w}_t-w_t^*\|^2 + \frac{534\beta\rho_{\max}^2(\nu+1)}{N\lambda_2(1-\delta)}\Big(1+\frac{1}{\lambda_2}\Big)^2(\|\overline{\theta}_t-\theta^*\|^2+\|\theta^*\|^2+R_{\max}^2)$$

$$+ \frac{15\beta\sigma_2^{L/4}}{M\lambda_2}\big(17\max_{m\in\mathcal{M}}(\|\theta_0^{(m)}\|+\|w_0^{(m)}\|+R_{\max})^2[1+\beta(2\rho_{\max}+3)]^{2t}\big)$$

$$+ \underbrace{\frac{24\alpha^2\rho_{\max}^2}{\beta\lambda_2^3}\mathbb{E}\Big[\Big\|\frac{1}{M}\sum_{m=1}^M\big(\widetilde{A}_t^{(m)}\theta_t^{(m)}+\widetilde{\widetilde{b}}_t^{(m)}+\widetilde{B}_t^{(m)}w_t^{(m)}\big)\Big\|^2\Big|\mathcal{F}_t\Big]}_{(V)},\tag{25}$$

where (i) uses the inequality that $\|a_1+a_2\|^2\le(1+\sigma)\|a_1\|^2+(1+\sigma^{-1})\|a_2\|^2$ for any $a_1,a_2\in\mathbb{R}^d$ and $\sigma>0$, (ii) uses eq. (24) as well as the notation that $w_t^*=-C^{-1}(A\overline{\theta}_t+b)$, (iii) uses $\beta\le\frac{1}{\lambda_2}$ (this implies that $\frac{6/(\beta\lambda_2)-1}{2[3/(\beta\lambda_2)-1]}\le\frac{5}{4}$) and eqs. (14), (39) and (43). The above term (V) can be upper bounded as follows.

$$(V) = \mathbb{E}\Big[\Big\|\frac{1}{M}\sum_{m=1}^M\big(\widetilde{A}_t^{(m)}\theta_t^{(m)}+\widetilde{\widetilde{b}}_t^{(m)}+\widetilde{B}_t^{(m)}w_t^{(m)}\big)\Big\|^2\Big|\mathcal{F}_t\Big]$$

$$\overset{(i)}{\le} 2\mathbb{E}\Big[\Big\|\frac{1}{M}\sum_{m=1}^M\big((\widetilde{A}_t^{(m)}-\widetilde{A}_t)\theta_t^{(m)}+(\widetilde{\widetilde{b}}_t^{(m)}-\widetilde{b}_t^{(m)})+(\widetilde{B}_t^{(m)}-\widetilde{B}_t)w_t^{(m)}\big)\Big\|^2\Big|\mathcal{F}_t\Big]$$

$$+ 2\mathbb{E}\Big[\Big\|\frac{1}{M}\sum_{m=1}^M\big(\widetilde{A}_t\theta_t^{(m)}+\widetilde{b}_t^{(m)}+\widetilde{B}_tw_t^{(m)}\big)\Big\|^2\Big|\mathcal{F}_t\Big]$$

$$\overset{(ii)}{\le} \frac{2}{M}\sum_{m=1}^M\mathbb{E}\big[\big\|\big((\widetilde{A}_t^{(m)}-\widetilde{A}_t)\theta_t^{(m)}+(\widetilde{\widetilde{b}}_t^{(m)}-\widetilde{b}_t^{(m)})+(\widetilde{B}_t^{(m)}-\widetilde{B}_t)w_t^{(m)}\big)\big\|^2\big|\mathcal{F}_t\big]+2\mathbb{E}\big(\big\|\widetilde{A}_t\overline{\theta}_t+\widetilde{\overline{b}}_t+\widetilde{B}_t\overline{w}_t\big\|^2\big|\mathcal{F}_t\big)$$

$$\overset{(iii)}{\le} \frac{6}{M}\sum_{m=1}^M\mathbb{E}\big[\|\widetilde{A}_t^{(m)}-\widetilde{A}_t\|^2\|\theta_t^{(m)}\|^2+\|\widetilde{\widetilde{b}}_t^{(m)}-\widetilde{b}_t^{(m)}\|^2+\|\widetilde{B}_t^{(m)}-\widetilde{B}_t\|^2\|w_t^{(m)}\|^2\big|\mathcal{F}_t\big]$$

$$+ 2\Big(40\rho_{\max}^2\Big(1+\frac{\rho_{\max}}{\lambda_2}\Big)^2\|\overline{\theta}_t-\theta^*\|^2+5\rho_{\max}^2\|\overline{w}_t-w_t^*\|^2+\frac{160\rho_{\max}^2(\nu+1)}{N(1-\delta)}\Big(1+\frac{\rho_{\max}}{\lambda_2}\Big)^2(\|\theta^*\|^2+R_{\max}^2)\Big)$$

$$\overset{(iv)}{\le} \frac{6\sigma_2^{L/4}}{M}\Big(20\max_{m\in\mathcal{M}}(\|\theta_0^{(m)}\|+\|w_0^{(m)}\|+R_{\max})^2[1+\beta(2\rho_{\max}+3)]^{2t}+R_{\max}^2\Big)$$

$$+ 2\Big(40\rho_{\max}^2\Big(1+\frac{\rho_{\max}}{\lambda_2}\Big)^2\|\overline{\theta}_t-\theta^*\|^2+5\rho_{\max}^2\|\overline{w}_t-w_t^*\|^2+\frac{160\rho_{\max}^2(\nu+1)}{N(1-\delta)}\Big(1+\frac{\rho_{\max}}{\lambda_2}\Big)^2(\|\theta^*\|^2+R_{\max}^2)\Big)$$

where (i) uses the inequality that $\|a_1+a_2\|^2\le2\|a_1\|^2+2\|a_2\|^2$ for any $a_1,a_2\in\mathbb{R}^d$, (ii) applies Jensen's inequality to the convex function $\|\cdot\|^2$, (iii) uses eq. (21) and the inequality that $\|a_1+a_2+a_3\|^2\le3\sum_{k=1}^3\|a_k\|^2$ for any $a_1,a_2,a_3\in\mathbb{R}^d$, (iv) uses eqs. (52), (53), (54) and (68). Substituting the above inequality into eq. (25) yields that

$$\mathbb{E}\big[\|\overline{w}_{t+1}-w_{t+1}^*\|^2\big|\mathcal{F}_t\big]$$

$$\le \Big(1-\frac{\beta\lambda_2}{6}+\frac{120\alpha^2\rho_{\max}^4}{\beta\lambda_2^3}\Big)\|\overline{w}_t-w_t^*\|^2+\Big[\frac{534\beta\rho_{\max}^2(\nu+1)}{N\lambda_2(1-\delta)}\Big(1+\frac{1}{\lambda_2}\Big)^2+\frac{1920\alpha^2\rho_{\max}^4}{\beta\lambda_2^3}\Big(1+\frac{\rho_{\max}}{\lambda_2}\Big)^2\Big]\|\overline{\theta}_t-\theta^*\|^2$$

$$+ \frac{3840\rho_{\max}^2(\nu+1)}{N(1-\delta)}\Big[\frac{\beta}{\lambda_2}\Big(1+\frac{1}{\lambda_2}\Big)^2+\frac{\alpha^2\rho_{\max}^2}{\beta\lambda_2^3}\Big(1+\frac{\rho_{\max}}{\lambda_2}\Big)^2\Big](\|\theta^*\|^2+R_{\max}^2)$$

$$+ \frac{255\sigma_2^{L/4}}{M\lambda_2}\Big(\beta + \frac{12\alpha^2\rho_{\max}^2}{\beta\lambda_2^2}\Big)\max_{m\in\mathcal{M}}(\|\theta_0^{(m)}\| + \|w_0^{(m)}\| + R_{\max})^2[1 + \beta(2\rho_{\max} + 3)]^{2t}. \tag{26}$$

$$\mathbb{E}\big[\|\overline{\theta}_{t+1} - \theta^*\|^2\big|\mathcal{F}_t\big]$$
$$\overset{(i)}{\leq} \Big(1 + \frac{1}{6/(\alpha\lambda_1) - 3}\Big)\mathbb{E}\big[\widetilde{\theta}_{t+1} - \theta^*\big|\mathcal{F}_t\big] + \Big(1 + \frac{6}{\alpha\lambda_1} - 3\Big)\mathbb{E}\big[\overline{\theta}_{t+1} - \widetilde{\theta}_{t+1}\big|\mathcal{F}_t\big]$$
$$\overset{(ii)}{\leq} \frac{6 - 2\alpha\lambda_1}{6 - 3\alpha\lambda_1}\Big[\Big(1 - \frac{\alpha\lambda_1}{2}\Big)\|\overline{\theta}_t - \theta^*\|^2 + \frac{13\alpha\rho_{\max}^2}{\lambda_1}\|\overline{w}_t - w_t^*\|^2$$
$$+ \frac{224\alpha\rho_{\max}^2(\nu + 1)}{N\lambda_1(1 - \delta)}\big(4\|\theta^*\|^2 + 2R_{\max}^2 + 1\big)\Big(1 + \frac{\rho_{\max}}{\lambda_2}\Big)^2\Big]$$
$$+ \frac{6}{\alpha\lambda_1}\mathbb{E}\Big[\Big\|\frac{\alpha}{M}\sum_{m=1}^{M}\big[(\widetilde{A}_t^{(m)} - \widetilde{A}_t)\theta_t^{(m)} + \widetilde{\widetilde{b}}_t^{(m)} - \widetilde{b}_t^{(m)} + (\widetilde{B}_t^{(m)} - \widetilde{B}_t)w_t^{(m)}]\Big\|^2\Big|\mathcal{F}_t\Big]$$
$$\overset{(iii)}{\leq} \Big(1 - \frac{\alpha\lambda_1}{3}\Big)\|\overline{\theta}_t - \theta^*\|^2 + \frac{18\alpha\rho_{\max}^2}{\lambda_1}\|\overline{w}_t - w_t^*\|^2 + \frac{300\alpha\rho_{\max}^2(\nu + 1)}{N\lambda_1(1 - \delta)}\big(4\|\theta^*\|^2 + 2R_{\max}^2 + 1\big)\Big(1 + \frac{\rho_{\max}}{\lambda_2}\Big)^2$$
$$+ \frac{6\alpha}{M\lambda_1}\sum_{m=1}^{M}\mathbb{E}\big[\|(\widetilde{A}_t^{(m)} - \widetilde{A}_t)\theta_t^{(m)} + \widetilde{\widetilde{b}}_t^{(m)} - \widetilde{b}_t^{(m)} + (\widetilde{B}_t^{(m)} - \widetilde{B}_t)w_t^{(m)}\|^2\big|\mathcal{F}_t\big]$$
$$\overset{(iv)}{\leq} \Big(1 - \frac{\alpha\lambda_1}{3}\Big)\|\overline{\theta}_t - \theta^*\|^2 + \frac{18\alpha\rho_{\max}^2}{\lambda_1}\|\overline{w}_t - w_t^*\|^2 + \frac{300\alpha\rho_{\max}^2(\nu + 1)}{N\lambda_1(1 - \delta)}\big(4\|\theta^*\|^2 + 2R_{\max}^2 + 1\big)\Big(1 + \frac{\rho_{\max}}{\lambda_2}\Big)^2$$
$$+ \frac{18\alpha\sigma_2^{L/4}}{M\lambda_1}\Big[20\max_{m\in\mathcal{M}}(\|\theta_0^{(m)}\| + \|w_0^{(m)}\| + R_{\max})^2[1 + \beta(2\rho_{\max} + 3)]^{2t} + R_{\max}^2\Big], \tag{27}$$

where (i) uses the inequality that $\|a_1 + a_2\|^2 \leq (1 + \sigma)\|a_1\|^2 + (1 + \sigma^{-1})\|a_2\|^2$ for any $a_1, a_2 \in \mathbb{R}^d$ and $\sigma > 0$, (ii) uses eqs. (14), (12) and (23), (iii) uses $\alpha \leq 1/\lambda_1$ and applies Jensen's inequality to the convex function $\|\cdot\|^2$, and (iv) uses the inequality that $\|a_1 + a_2 + a_3\|^2 \leq 3\sum_{k=1}^{3}\|a_k\|^2$ for any $a_1, a_2, a_3 \in \mathbb{R}^d$ and then uses eqs. (52)-(54).

Taking expectation on both sides of eqs. (26) and (27) and summing up the two inequalities yields that

$$\mathbb{E}(\|\overline{\theta}_{t+1} - \theta^*\|^2) + \mathbb{E}(\|\overline{w}_{t+1} - w_{t+1}^*\|^2)$$
$$\overset{(i)}{\leq} \Big(1 - \frac{\beta\lambda_2}{6} + \frac{120\alpha^2\rho_{\max}^4}{\beta\lambda_2^3} + \frac{18\alpha\rho_{\max}^2}{\lambda_1}\Big)\mathbb{E}(\|\overline{w}_t - w_t^*\|^2)$$
$$+ \Big[1 - \frac{\alpha\lambda_1}{3} + \frac{534\beta\rho_{\max}^2(\nu + 1)}{N\lambda_2(1 - \delta)}\Big(1 + \frac{1}{\lambda_2}\Big)^2 + \frac{1920\alpha^2\rho_{\max}^4}{\beta\lambda_2^3}\Big(1 + \frac{\rho_{\max}}{\lambda_2}\Big)^2\Big]\mathbb{E}(\|\overline{\theta}_t - \theta^*\|^2)$$
$$+ \frac{3840\rho_{\max}^2(\nu + 1)}{N(1 - \delta)}\Big[\frac{\beta}{\lambda_2}\Big(1 + \frac{1}{\lambda_2}\Big)^2 + \frac{\alpha^2\rho_{\max}^2}{\beta\lambda_2^3}\Big(1 + \frac{\rho_{\max}}{\lambda_2}\Big)^2 + \frac{\alpha}{\lambda_1}\Big](\|\theta^*\|^2 + R_{\max}^2 + 1)$$
$$+ \frac{255\sigma_2^{L/4}}{M\lambda_2}\Big(\beta + \frac{12\alpha^2\rho_{\max}^2}{\beta\lambda_2^2} + \frac{\alpha\lambda_2}{\lambda_1}\Big)\max_{m\in\mathcal{M}}(\|\theta_0^{(m)}\| + \|w_0^{(m)}\| + R_{\max})^2[1 + \beta(2\rho_{\max} + 3)]^{2t}$$
$$\overset{(i)}{\leq} \Big(1 - \frac{\alpha\lambda_1}{6}\Big)\big[\mathbb{E}(\|\overline{\theta}_t - \theta^*\|^2) + \mathbb{E}(\|\overline{w}_t - w_t^*\|^2)\big] + \frac{6000\beta\rho_{\max}^2(\nu + 1)}{N\lambda_2(1 - \delta)}\Big(1 + \frac{\rho_{\max}}{\lambda_2}\Big)^2(\|\theta^*\|^2 + R_{\max}^2 + 1)$$
$$+ \frac{574\beta\sigma_2^{L/4}}{M\lambda_2}\max_{m\in\mathcal{M}}(\|\theta_0^{(m)}\| + \|w_0^{(m)}\| + R_{\max})^2[1 + \beta(2\rho_{\max} + 3)]^{2t},$$

where (i) uses the conditions that $N \geq \frac{6408\beta\rho_{\max}^2(\nu+1)}{\alpha\lambda_1\lambda_2(1-\delta)}\big(1 + \frac{1}{\lambda_2}\big)^2$, $\alpha \leq \min\big(\frac{\beta\lambda_1\lambda_2^3}{23040\rho_{\max}^4}, \frac{\beta\lambda_2^2}{53\rho_{\max}^2}, \frac{\beta\lambda_1\lambda_2}{432\rho_{\max}^2}, \frac{\beta\lambda_2}{2\lambda_1}, \frac{\beta\lambda_1}{2\lambda_2}, \frac{\beta\lambda_2}{4\rho_{\max}}\big)$. Iterating the inequality above yields the following convergence rate of the optimization error $\mathbb{E}\big[\|\overline{w}_{t+1} - w_{t+1}^*\|^2\big|\mathcal{F}_t\big]$ and $\mathbb{E}\big[\|\overline{\theta}_{t+1} - \theta^*\|^2\big|\mathcal{F}_t\big]$

$$\mathbb{E}\big(\|\overline{\theta}_T - \theta^*\|^2 + \|\overline{w}_T - w^*\|^2\big)$$

$$\leq \Big(1 - \frac{\alpha\lambda_1}{6}\Big)^T \big(\|\overline{\theta}_0 - \theta^*\|^2 + \|\overline{w}_0 - w_0^*\|^2\big)$$

$$+ \sum_{t=0}^{T-1}\Big(1 - \frac{\alpha\lambda_1}{6}\Big)^{T-1-t}\Big[\frac{6000\beta\rho_{\max}^2(\nu+1)}{N\lambda_2(1-\delta)}\Big(1 + \frac{\rho_{\max}}{\lambda_2}\Big)^2\big(\|\theta^*\|^2 + R_{\max}^2 + 1\big)$$

$$+ \frac{574\beta\sigma_2^{L/4}}{M\lambda_2}\max_{m\in\mathcal{M}}(\|\theta_0^{(m)}\| + \|w_0^{(m)}\| + R_{\max})^2[1 + \beta(2\rho_{\max}+3)]^{2t}\Big]$$

$$\overset{(i)}{\leq}\Big(1 - \frac{\alpha\lambda_1}{6}\Big)^T\big(\|\overline{\theta}_0 - \theta^*\|^2 + \|\overline{w}_0 - w_0^*\|^2\big) + \frac{36000\beta\rho_{\max}^2(\nu+1)}{\alpha N\lambda_1\lambda_2(1-\delta)}\Big(1 + \frac{\rho_{\max}}{\lambda_2}\Big)^2\big(\|\theta^*\|^2 + R_{\max}^2 + 1\big)$$

$$+ \frac{574\beta\sigma_2^{L/4}2^T}{M\lambda_2}\max_{m\in\mathcal{M}}(\|\theta_0^{(m)}\| + \|w_0^{(m)}\| + R_{\max})^2$$

$$= \Big(1 - \frac{\alpha\lambda_1}{6}\Big)^T\big(\|\overline{\theta}_0 - \theta^*\|^2 + \|\overline{w}_0 - w_0^*\|^2\big) + \mathcal{O}\Big(\frac{\beta}{N\alpha} + \frac{\beta\sigma_2^{L/4}2^T}{M}\Big), \tag{28}$$

where (i) uses the conditions that $\alpha \leq \frac{1}{\lambda_1}$ and $\beta \leq \frac{2}{5(2\rho_{\max}+3)}$ which respectively imply that $1 - \frac{\alpha\lambda_1}{6} \geq \frac{5}{6}$ and that $1 + \beta(2\rho_{\max}+3) \leq \sqrt{2}$. This proves eq. (10).

**Step 3: Proof of consensus error (11).** Note that the local model averaging iterations can be rewritten into the matrix-vector form as $\Theta_{t+1} = U\Theta_t$ where $T \leq t \leq T + T'$ and $\Theta_t \overset{\triangle}{=} [\theta_t^{(1)}; \theta_t^{(2)}; \ldots; \theta_t^{(M)}]^\top$. Hence, it can be derived from Lemma C.3 that

$$\|\Delta\Theta_{T+T'}\|_F = \|\Delta U^{T'}\Theta_T\|_F = \|U^{T'}\Delta\Theta_T\|_F \leq \sigma_2^{T'}\|\Delta\Theta_T\|_F. \tag{29}$$

Hence, we only need to obtain an upper bound of the initial consensus error $\mathbb{E}\|\Delta\Theta_T\|^2$. Subtracting eq. (14) from the local update rule yields that for any $0 \leq t \leq T - 1$,

$$\theta_{t+1}^{(m)} - \overline{\theta}_{t+1} = \sum_{m'\in\mathcal{N}_m} U_{m,m'}(\theta_t^{(m')} - \overline{\theta}_t) + \frac{M-1}{M}\alpha\big(\widetilde{A}_t^{(m)}\theta_t^{(m)} + \widetilde{\overline{b}}_t^{(m)} + \widetilde{B}_t^{(m)}w_t^{(m)}\big)$$

$$- \frac{\alpha}{M}\sum_{m'=1,m'\neq m}^{M}\big(\widetilde{A}_t^{(m')}\theta_t^{(m')} + \widetilde{\overline{b}}_t^{(m')} + \widetilde{B}_t^{(m')}w_t^{(m')}\big).$$

This can be rewritten into the following matrix-vector form,

$$\Delta\Theta_{t+1} = U\Delta\Theta_t + [h_1; h_2; \ldots; h_M]^\top,$$

where $h_m \overset{\triangle}{=} \frac{M-1}{M}\alpha\big(\widetilde{A}_t^{(m)}\theta_t^{(m)} + \widetilde{\overline{b}}_t^{(m)} + \widetilde{B}_t^{(m)}w_t^{(m)}\big) - \frac{\alpha}{M}\sum_{m'=1,m'\neq m}^{M}\big(\widetilde{A}_t^{(m')}\theta_t^{(m')} + \widetilde{\overline{b}}_t^{(m')} + \widetilde{B}_t^{(m')}w_t^{(m')}\big).$
The item 2 of Lemma C.3 implies that for any $0 \leq t \leq T - 1$,

$$\|\Delta\Theta_{t+1}\|_F \leq \sigma_2\|\Delta\Theta_t\|_F + \sqrt{\sum_{m=1}^{M}\|h_m\|^2} \leq \sigma_2\|\Delta\Theta_t\|_F + \sum_{m=1}^{M}\|h_m\|. \tag{30}$$

Then, using eqs. (55)-(57) yields that

$$\sum_{m=1}^{M}\|h_m\| \leq \frac{M-1}{M}(2\alpha)(2\rho_{\max}+2)\sum_{m=1}^{M}\big(\|\theta_t^{(m)}\| + R_{\max} + \|w_t^{(m)}\|\big)$$

$$\overset{(i)}{\leq} 4\alpha(\rho_{\max}+1)\sum_{m=1}^{M}\big(\|\theta_t^{(m)} - \overline{\theta}_t\| + \|w_t^{(m)} - \overline{w}_t\| + \|\overline{\theta}_t - \theta^*\| + \|\overline{w}_t - w_t^*\| + \|\theta^*\| + \|C^{-1}(A\overline{\theta}_t + b)\|\big)$$

$$\overset{(ii)}{\leq} 4\alpha(\rho_{\max}+1)\Big(\sum_{m=1}^{M}\big(\|\theta_t^{(m)} - \overline{\theta}_t\| + \|w_t^{(m)} - \overline{w}_t\|\big)$$

$$+ M\Big(1+\frac{2\rho_{\max}}{\lambda_2}\Big)\|\overline{\theta}_t - \theta^*\| + M\|\overline{w}_t - w_t^*\| + \frac{M\rho_{\max}}{\lambda_2}(2\|\theta^*\| + R_{\max})\Big),$$

where (i) uses the notations that $w_t^* = -C^{-1}(A\overline{\theta}_t + b)$, (ii) uses eqs. (39), (42) and (43). Hence, we obtain that

$$\mathbb{E}(\|\Delta\Theta_{t+1}\|_F^2) \overset{(i)}{\leq} \Big(1 + \frac{\sigma_2^{-2}-1}{2}\Big)\sigma_2^2 \mathbb{E}(\|\Delta\Theta_t\|_F^2) + \Big(1 + \frac{2}{\sigma_2^{-2}-1}\Big)\mathbb{E}\Big[\Big(\sum_{m=1}^{M}\|h_m\|\Big)^2\Big],$$

$$\overset{(ii)}{\leq} \frac{1+\sigma_2^2}{2}\mathbb{E}(\|\Delta\Theta_t\|_F^2) + \frac{48\alpha^2(1+\sigma_2^2)}{1-\sigma_2^2}(\rho_{\max}+1)^2\mathbb{E}\Big[2M\sum_{m=1}^{M}\big(\|\theta_t^{(m)} - \overline{\theta}_t\|^2 + \|w_t^{(m)} - \overline{w}_t\|^2\big)$$

$$+ M^2\Big(1+\frac{2\rho_{\max}}{\lambda_2}\Big)^2\big(\|\overline{\theta}_t - \theta^*\|^2 + \|\overline{w}_t - w_t^*\|^2\big) + \frac{4M^2\rho_{\max}^2}{\lambda_2^2}(\|\theta^*\| + R_{\max})^2\Big], \qquad (31)$$

where (i) uses eq. (30) and the fact that $(u+v)^2 \leq (1+\sigma)u^2 + (1+\sigma^{-1})v^2$ for any $u, v, \sigma \geq 0$, (ii) uses $(\sum_{i=1}^{n} q_i)^2 \leq n\sum_{i=1}^{n} q_i^2$ for any $q_i \in \mathbb{R}$ and $n \in \mathbb{N}^+$. Similarly, we obtain from the update rule of $W_t = [w_t^{(1)}; w_t^{(2)}; \ldots; w_t^{(M)}]^\top \in \mathbb{R}^{M\times d}$ and eq. (15) that

$$\mathbb{E}(\|\Delta W_{t+1}\|_F^2) \leq \frac{1+\sigma_2^2}{2}\mathbb{E}\big(\|\Delta W_t\|_F^2\big) + \frac{48\alpha^2(1+\sigma_2^2)}{1-\sigma_2^2}(\rho_{\max}+1)^2\mathbb{E}\Big[2M\sum_{m=1}^{M}\big(\|\theta_t^{(m)} - \overline{\theta}_t\|^2 + \|w_t^{(m)} - \overline{w}_t\|^2\big)$$

$$+ M^2\Big(1+\frac{2\rho_{\max}}{\lambda_2}\Big)^2\big(\|\overline{\theta}_t - \theta^*\|^2 + \|\overline{w}_t - w_t^*\|^2\big) + \frac{4M^2\rho_{\max}^2}{\lambda_2^2}(\|\theta^*\| + R_{\max})^2\Big], \qquad (32)$$

Summing up eqs. (31) and (32) yields that

$$\mathbb{E}(\|\Delta\Theta_{t+1}\|_F^2 + \|\Delta W_{t+1}\|_F^2)$$

$$\overset{(i)}{\leq} \frac{1+\sigma_2^2}{2}\mathbb{E}\big(\|\Delta\Theta_t\|_F^2 + \|\Delta W_t\|_F^2\big) + \frac{96\alpha^2(1+\sigma_2^2)}{1-\sigma_2^2}(\rho_{\max}+1)^2\mathbb{E}\Big[2M\big(\|\Delta\Theta_t\|_F^2 + \|\Delta W_t\|_F^2\big)$$

$$+ M^2\Big(1+\frac{2\rho_{\max}}{\lambda_2}\Big)^2\Big(\Big(1-\frac{\alpha\lambda_1}{6}\Big)^t\big(\|\overline{\theta}_0 - \theta^*\|^2 + \|\overline{w}_0 - w_0^*\|^2\big)$$

$$+ \frac{36000\beta\rho_{\max}^2(\nu+1)}{\alpha N\lambda_1\lambda_2(1-\delta)}\Big(1+\frac{\rho_{\max}}{\lambda_2}\Big)^2\big(\|\theta^*\|^2 + R_{\max}^2 + 1\big) + \frac{574\beta\sigma_2^{L/4}2^t}{M\lambda_2}\max_{m\in\mathcal{M}}(\|\theta_0^{(m)}\| + \|w_0^{(m)}\| + R_{\max})^2\Big)$$

$$+ \frac{4M^2\rho_{\max}^2}{\lambda_2^2}(\|\theta^*\| + R_{\max})^2\Big]$$

$$\overset{(ii)}{\leq} \frac{2+\sigma_2^2}{3}\mathbb{E}\big(\|\Delta\Theta_t\|_F^2 + \|\Delta W_t\|_F^2\big) + \frac{192M^2\alpha^2}{1-\sigma_2^2}(\rho_{\max}+1)^2\Big(1+\frac{2\rho_{\max}}{\lambda_2}\Big)^2\Big[\big(\|\overline{\theta}_0 - \theta^*\|^2 + \|\overline{w}_0 - w_0^*\|^2\big)$$

$$+ \Big(\frac{282\beta}{\alpha\lambda_2} + \frac{8M^2\rho_{\max}^2}{\lambda_2^2}\Big)\big(\|\theta^*\|^2 + R_{\max}^2 + 1\big) + \frac{574\beta\sigma_2^{L/4}2^t}{M\lambda_2}\max_{m\in\mathcal{M}}(\|\theta_0^{(m)}\| + \|w_0^{(m)}\| + R_{\max})^2\Big)\Big], \qquad (33)$$

where (i) uses eq. (28), and (ii) uses $(\|\theta^*\| + R_{\max})^2 \leq 2\|\theta^*\|^2 + 2R_{\max}^2$, $\alpha \leq \frac{1-\sigma_2}{50\sqrt{M}(\rho_{\max}+1)}$ and $N \geq \frac{128\rho_{\max}^2(\nu+1)}{\lambda_1(1-\delta)}\Big(1+\frac{\rho_{\max}}{\lambda_2}\Big)^2$.

Iterating eq. (33) yields the following convergence rate of the initial consensus error $\mathbb{E}\big(\|\Delta\Theta_T\|_F^2\big)$.

$$\mathbb{E}\big(\|\Delta\Theta_T\|_F^2\big) \leq \mathbb{E}\big(\|\Delta\Theta_T\|_F^2 + \|\Delta W_T\|_F^2\big)$$

$$\leq \Big(\frac{2+\sigma_2^2}{3}\Big)^T\big(\|\Delta\Theta_0\|_F^2 + \|\Delta W_0\|_F^2\big) + \frac{192M^2\alpha^2}{1-\sigma_2^2}(\rho_{\max}+1)^2\Big(1+\frac{2\rho_{\max}}{\lambda_2}\Big)^2$$

$$+ \Big[\frac{3}{1-\sigma_2}\Big(\|\overline{\theta}_0 - \theta^*\|^2\|\overline{w}_0 - w_0^*\|^2 + \Big(\frac{282\beta}{\alpha\lambda_2} + \frac{8M^2\rho_{\max}^2}{\lambda_2^2}\Big)\big(\|\theta^*\|^2 + R_{\max}^2 + 1\big)\Big)$$

$$+ \frac{574\beta\sigma_2^{L/4}2^T}{M\lambda_2}\max_{m\in\mathcal{M}}(\|\theta_0^{(m)}\| + \|w_0^{(m)}\| + R_{\max})^2\Big]$$

$$\leq \|\Delta\Theta_0\|_F^2 + \|\Delta W_0\|_F^2 + \mathcal{O}\Big[\frac{M^2\alpha^2}{(1-\sigma_2)^2}\Big(\frac{\beta}{\alpha}+M^2\Big) + \frac{M\beta\alpha^2\sigma_2^{L/4}2^T}{1-\sigma_2}\Big]$$

$$\overset{(i)}{\leq} \mathcal{O}\Big(1 + \frac{M^4\beta\alpha}{(1-\sigma_2)^2} + \frac{M\beta\alpha\sigma_2^{L/4}2^T}{1-\sigma_2}\Big), \tag{34}$$

where (i) uses $\alpha \leq \frac{\beta}{2\rho_{\max}+2} = \mathcal{O}(\beta)$. Substituting the above inequality into eq. (29) proves eq. (11).

**Hyperparameter choice and complexities.** To summarize, the following conditions of the hyperparameters are used in the proof of Theorem 1, including those required by Corollary 2 and lemma C.4.

$$\alpha \leq \min\Big(\frac{\beta}{2\rho_{\max}+2}, \frac{\lambda_1}{40\rho_{\max}^2}\Big(1+\frac{\rho_{\max}}{\lambda_2}\Big)^{-2}, \frac{1}{\lambda_1}, \frac{\beta\lambda_2}{4\rho_{\max}}, \frac{\beta\lambda_1\lambda_2^3}{23040\rho_{\max}^4}, \frac{\beta\lambda_2^2}{53\rho_{\max}^2}, \frac{\beta\lambda_1\lambda_2}{432\rho_{\max}^2},$$

$$\frac{1-\sigma_2}{50\sqrt{M}(\rho_{\max}+1)}, \frac{\beta\lambda_2}{2\lambda_1}, \frac{\beta\lambda_1}{2\lambda_2}\Big) = \min\{\mathcal{O}(\beta), \mathcal{O}(M^{-1/2}(1-\sigma_2))\} \tag{35}$$

$$\beta \leq \min\Big(\frac{1}{\lambda_2}, \frac{2}{5(2\rho_{\max}+3)}\Big) = \mathcal{O}(1) \tag{36}$$

$$N \geq \max\Big(\frac{8\nu\rho_{\max}}{\lambda_2(1-\delta)}, \frac{6408\beta\rho_{\max}^2(\nu+1)}{\alpha\lambda_1\lambda_2(1-\delta)}\Big(1+\frac{1}{\lambda_2}\Big)^2, \frac{128\rho_{\max}^2(\nu+1)}{\lambda_1(1-\delta)}\Big(1+\frac{\rho_{\max}}{\lambda_2}\Big)^2\Big) = \max\{\mathcal{O}(1), \mathcal{O}(\beta/\alpha)\} \tag{37}$$

$$L \geq \frac{12\ln M + (8M+10)\ln\rho_{\max}}{\ln\sigma_2^{-1}} = \mathcal{O}\Big(\frac{M}{1-\sigma_2}\Big) \tag{38}$$

Under the above conditions, we choose the following hyperparameter values.

$$\alpha = \mathcal{O}(M^{-1/2}(1-\sigma_2)), \beta = \mathcal{O}(1)$$

$$T = \frac{6}{\alpha\lambda_1}\ln\epsilon^{-1} = \mathcal{O}\Big(\frac{\sqrt{M}\ln\epsilon^{-1}}{1-\sigma_2}\Big)$$

$$N = \frac{\beta}{\alpha\epsilon} = \mathcal{O}\Big(\frac{\sqrt{M}}{\epsilon(1-\sigma_2)}\Big)$$

$$L = \frac{4}{\ln\sigma_2^{-1}}\Big(\ln\Big(\frac{\beta}{M\epsilon}\Big)+T\ln 2\Big) + \frac{12\ln M + (8M+10)\ln\rho_{\max}}{\ln\sigma_2^{-1}} = \mathcal{O}\Big(\frac{\sqrt{M}\ln\epsilon^{-1}}{(1-\sigma_2)^2} + \frac{M}{1-\sigma_2}\Big) \leq \mathcal{O}\Big(\frac{M\ln\epsilon^{-1}}{(1-\sigma_2)^2}\Big)$$

$$T' = \frac{1}{\ln\sigma_2^{-1}}\ln\Big(\epsilon^{-1}\mathcal{O}\Big(1 + \frac{M^4\beta\alpha}{(1-\sigma_2)^2} + \frac{M\beta\alpha\sigma_2^{L/4}2^T}{1-\sigma_2}\Big)\Big)$$

$$= \frac{1}{\ln\sigma_2^{-1}}\ln\Big(\epsilon^{-1}\mathcal{O}\Big(1 + \frac{M^{3.5}}{1-\sigma_2} + \frac{\alpha M^2\epsilon}{1-\sigma_2}\Big)\Big) = \mathcal{O}\Big(\frac{1}{1-\sigma_2}\ln\Big(\frac{M}{\epsilon(1-\sigma_2)}\Big)\Big).$$

Substituting these hyperparameters into eqs. (10) and (11) implies $\mathbb{E}(\|\bar{\theta}_T - \theta^*\|^2), \mathbb{E}(\|\theta_{T+T'}^{(m)} - \bar{\theta}_T\|^2) \leq \mathcal{O}(\epsilon)$, so $\mathbb{E}(\|\theta_{T+T'}^{(m)} - \theta^*\|^2) \leq 2\mathbb{E}(\|\theta_{T+T'}^{(m)} - \bar{\theta}_T\|^2) + 2\mathbb{E}(\|\bar{\theta}_T - \theta^*\|^2) \leq \mathcal{O}(\epsilon)$. Therefore, the overall communication complexity for synchronizing $\theta_t^{(m)}$ is $T + T' = \mathcal{O}\big(\frac{\sqrt{M}\ln\epsilon^{-1}}{1-\sigma_2}\big)$, and the total sample complexity is $NT = \mathcal{O}\big(\frac{M\ln\epsilon^{-1}}{\epsilon(1-\sigma_2)^2}\big)$.

## C Supporting Lemmas

In this section, we prove some supporting lemmas that are used throughout the analysis of Algorithm 1.

**Lemma C.1.** *Regarding the terms defined in Appendix A, their norms have the following upper bounds.*

$$\|A_i\|_F, \|A_i^{(m)}\|_F, \|\widetilde{A}_t\|_F, \|A\|_F \leq 2\rho_{\max}, \tag{39}$$

$$\|B_i\|_F, \|B_i^{(m)}\|_F, \|\widetilde{B}_t\|_F, \|B\|_F \leq \rho_{\max}, \tag{40}$$

$$\|C_i\|_F, \|\widetilde{C}_t\|_F, \|C\|_F \leq 1, \tag{41}$$

$$\|b_i^{(m)}\|, \|\widetilde{b}_t^{(m)}\|, \|\overline{b}_i\|, \|\overline{\overline{b}}_t\|, \|b\| \leq \rho_{\max} R_{\max}, \tag{42}$$

$$\|C^{-1}\| = \lambda_2^{-1}. \tag{43}$$

*Proof.* Consider any two vectors $u, v \in \mathbb{R}^d$, we have that $\|uv^\top\|_F = \sqrt{\mathrm{tr}(vu^\top uv^\top)} = \|u\|\|v\|$. Therefore, by Assumption 3, we obtain that

$$\begin{aligned}
\|A_i\|_F &\leq \rho_i \|\phi(s_i)\| \|\gamma\phi(s_{i+1}) - \phi(s_i)\| \leq \rho_{\max}\big[\gamma\|\phi(s_{i+1})\| + \|\phi(s_i)\|\big] \leq 2\rho_{\max}, \\
\|B_i\|_F &\leq \gamma\rho_i \|\phi(s_{i+1})\| \|\phi(s_i)\| \leq \rho_{\max}, \\
\|C_i\|_F &\leq \|\phi(s_i)\|^2 \leq 1, \\
\|b_i^{(m)}\| &\leq \rho_i R_i^{(m)} \|\phi(s_i)\| \leq \rho_{\max} R_{\max}.
\end{aligned}$$

The proof for $\|A_i^{(m)}\|_F$, $\|\widetilde{b}_t^{(m)}\|$, etc. is similar.

On the other hand, by Jensen's inequality, we obtain that

$$\|A\|_F = \|\mathbb{E}_{\pi_b}[A_i]\|_F \leq \mathbb{E}_{\pi_b}\|A_i\|_F \leq 2\rho_{\max}, \quad \|\widetilde{A}_t\|_F \leq \frac{1}{N} \sum_{i=tN}^{(t+1)N-1} \|A_i\|_F \leq 2\rho_{\max}.$$

The proof for the other remaining matrices in eqs. (39)-(42) is similar by using the Jensen's inequality. Finally, we prove eq. (43). Note that $-C = \mathbb{E}_{\pi_b}\big(\phi(s_i)\phi(s_i)\big) \succ 0$ with $\lambda_{\min}(-C) = -\lambda_{\max}(C) = \lambda_2$. Hence, $\|C^{-1}\| = \lambda_{\max}(-C^{-1}) = \lambda_{\min}^{-1}(-C) = \lambda_2^{-1}$. $\qquad\square$

**Lemma C.2.** *Suppose the MDP trajectory $\{s_i, a_i\}_{i \geq 0}$ is generated following a behavioral policy $\pi_b$ where $a_t \triangleq \{a_t^{(m)}\}_m$. For any deterministic mappings $Y : \mathcal{S} \times \mathcal{A}_1 \times \ldots \times \mathcal{A}_M \times \mathcal{S} \to \mathbb{R}^{p \times q}$ such that $\|Y(s, a, s')\|_F \leq C_y, \forall s, s' \in \mathcal{S}, a^{(m)} \in \mathcal{A}_m$ where $a = \{a^{(m)}\}_m$, we have*

$$\left\|\mathbb{E}\Big[\frac{1}{N} \sum_{i=tN}^{(t+1)N-1} Y(s_i, a_i, s_{i+1})\Big|\mathcal{F}_t\Big] - \overline{Y}\right\| \leq \frac{2\nu C_y}{N(1-\delta)},$$

$$\mathbb{E}\Big[\Big\|\frac{1}{N} \sum_{i=tN}^{(t+1)N-1} Y(s_i, a_i, s_{i+1}) - \overline{Y}\Big\|_F^2 \Big|\mathcal{F}_t\Big] \leq \frac{8C_y^2(\nu+1)}{N(1-\delta)},$$

*where $\overline{Y} = \mathbb{E}Y(s_i, a_i, s_{i+1})$.*

**Note:** A simplified version of the above lemma has been proposed and proved in Xu et al. (2020a), where $a_i$ and $s_{i+1}$ are omitted in the above inequality. We add $a_i$ and $s_{i+1}$ so that this lemma can be better applied to the quantities $A_i$, $B_i$, $C_i$ and $b_i^{(m)}$ which rely on $s_i$ as well as $a_i$ and $s_{i+1}$. The proof logic is very similar to that of Xu et al. (2020a) and thus omitted here.

**Corollary 1.** *Regarding the terms defined in Appendix A, they have the following upper bounds.*

$$\mathbb{E}\big[\|\widetilde{C}_t - C\|_F \big|\mathcal{F}_t\big] \leq \frac{2\nu\rho_{\max}}{N(1-\delta)} \tag{44}$$

$$\mathbb{E}\big[\big\|\widetilde{B}_t - B\big\|_F^2 \big|\mathcal{F}_t\big] \leq \frac{8\rho_{\max}^2(\nu+1)}{N(1-\delta)} \tag{45}$$

$$\mathbb{E}\big[\big\|\overline{\overline{b}}_t - b\big\|^2 \big|\mathcal{F}_t\big] \leq \frac{8\rho_{\max}^2 R_{\max}^2(\nu+1)}{N(1-\delta)} \tag{46}$$

$$\mathbb{E}\big[\big\|\widetilde{A}_t - \widetilde{C}_t C^{-1} A\big\|_F^2 \big|\mathcal{F}_t\big] \leq \frac{32\rho_{\max}^2(\nu+1)}{N(1-\delta)}\Big(1 + \frac{1}{\lambda_2}\Big)^2 \tag{47}$$

$$\mathbb{E}\big[\big\|\overline{\overline{b}}_t - \widetilde{C}_t C^{-1} b\big\|^2 \big|\mathcal{F}_t\big] \leq \frac{8\rho_{\max}^2 R_{\max}^2(\nu+1)}{N(1-\delta)}\Big(1 + \frac{1}{\lambda_2}\Big)^2 \tag{48}$$

$$\mathbb{E}\big[\big\|\widetilde{A}_t - \widetilde{B}_t C^{-1} A - A^\top C^{-1} A\big\|_F^2 \big|\mathcal{F}_t\big] \leq \frac{32\rho_{\max}^2(\nu+1)}{N(1-\delta)}\Big(1 + \frac{\rho_{\max}}{\lambda_2}\Big)^2, \tag{49}$$

*Proof.* Let $Y(s, a, s') = -\gamma\rho(s, a)\phi(s')\phi(s)^\top$ in Lemma C.2. Then it can be checked that $Y(s_t, a_t, s_{t+1}) = B_t, C_y = \rho_{\max}, \frac{1}{N}\sum_{i=tN}^{(t+1)N-1} Y(s_i, a_i, s_{i+1}) = \widetilde{B}_t$, and $\overline{Y} = \mathbb{E}_{\pi_b} Y(s_i, a_i, s_{i+1}) = B$.

Applying Lemma C.2 to these equations proves eq. (45). The eqs. (44) and (46) can be proved in a similar way.

Let $Y(s, a, s') = \rho(s, a)\phi(s)[\gamma\phi(s') - \phi(s)]^\top + \gamma\rho(s, a)\phi(s')\phi(s)^\top C^{-1}A$. Then, it can be checked that $Y(s_i, a_i, s_{i+1}) = A_i - B_i C^{-1}A$, $\frac{1}{N}\sum_{i=tN}^{(t+1)N-1} Y(s_i, a_i, s_{i+1}) = \widetilde{A}_t - \widetilde{B}_t C^{-1}A$. Moreover,

$$\|Y(s, a, s')\|_F \leq \rho_{\max}(\gamma + 1) + \gamma\rho_{\max}\|C^{-1}\|\|A\| \leq 2\rho_{\max} + \rho_{\max}(\lambda_2^{-1})(2\rho_{\max}) = 2\rho_{\max}(1 + \rho_{\max}/\lambda_2) := C_y,$$
$$\overline{Y} = \mathbb{E}_{\pi_b} Y(s_i, a_i, s_{i+1}) = A - BC^{-1}A = A^\top C^{-1}A.$$

Applying Lemma C.2 to these equations proves eq. (49). The equations (47) and (48) can be proved in a similar way. $\qquad\square$

**Lemma C.3.** *The doubly stochastic matrix $U$ and the difference matrix $\Delta = I - \frac{1}{M}\mathbf{1}\mathbf{1}^\top$ have the following properties:*

1. $\Delta U = U\Delta = U - \frac{1}{M}\mathbf{1}\mathbf{1}^\top$

2. *For any $x \in \mathbb{R}^M$ and $n \in \mathbb{N}^+$, $\|U^n \Delta x\| \leq \sigma_2^n\|\Delta x\|$ ($\sigma_2$ is the second largest singular value of $U$). Hence, for any $H \in \mathbb{R}^{M\times M}$, $\|U^n \Delta H\|_F \leq \sigma_2^n\|\Delta H\|_F$*

*Proof.* The first item can be proved by the following two equalities.

$$\Delta U = \left(I - \frac{1}{M}\mathbf{1}\mathbf{1}^\top\right)U = U - \frac{1}{M}\mathbf{1}\mathbf{1}^\top U = U - \frac{1}{M}\mathbf{1}\mathbf{1}^\top$$
$$U\Delta = U\left(I - \frac{1}{d}\mathbf{1}\mathbf{1}^\top\right) = U - \frac{1}{M}U\mathbf{1}\mathbf{1}^\top = U - \frac{1}{M}\mathbf{1}\mathbf{1}^\top$$

The proof of the item 2 follows from the claim in page 3 of Qu & Li (2017) that

$$\|Ux - \mathbf{1}\overline{x}\| \leq \sigma\|x - \mathbf{1}\overline{x}\|, \tag{50}$$

where we replace their $W \in \mathbb{R}^{n\times n}$ and vector $\omega \in \mathbb{R}^n$ into our $U \in \mathbb{R}^{M\times M}$ and vector $H \in \mathbb{R}^M$ respectively, $\overline{x} = \frac{1}{M}\mathbf{1}^\top x$ and $\sigma$ is the largest singular value of $U - \frac{1}{M}\mathbf{1}\mathbf{1}^\top$. We first prove that $\sigma$ is also the second largest singular value of $U$, i.e., $\sigma = \sigma_2$.

Consider the singular value decomposition of the doubly stochastic matrix $U = Q^\top D\widetilde{Q}$, where matrices $Q, \widetilde{Q}$ are unitary and matrix $D = \text{diag}(1, \sigma_2, \sigma_3, \ldots, \sigma_M)$ is diagonal with $1 > \sigma_2 \geq \sigma_3 \geq \sigma_M \geq 0$. Note that $\mathbf{1} = U\mathbf{1} = Q^\top D\widetilde{Q}\mathbf{1}$ which implies that $Q\mathbf{1} = D\widetilde{Q}\mathbf{1}$. Similarly, $\mathbf{1} = U^\top\mathbf{1} = \widetilde{Q}^\top DQ\mathbf{1} \Rightarrow \widetilde{Q}\mathbf{1} = DQ\mathbf{1}$. Combining the above two results, we conclude that

$$Q\mathbf{1} = D\widetilde{Q}\mathbf{1} = DDU\mathbf{1} = D^2 U\mathbf{1},$$

that is, $(I - D^2)Q\mathbf{1} = \mathbf{0}$. Since $I - D^2$ is a diagonal matrix where the first diagonal entry is zero but the rest diagonal entries are strictly positive, it must hold that all the entries of $Q\mathbf{1}$ are zero except for its first entry, i.e., $Q\mathbf{1} = \alpha e_1$ where $\alpha \in \mathbb{R}$ is the first entry of $Q\mathbf{1}$ and $e_1 = (1, 0, 0, \ldots, 0)$ is a basis vector. Hence, we conclude that $\widetilde{Q}\mathbf{1} = DQ\mathbf{1} = \alpha De_1 = \alpha e_1$. Taking the norm of both sides yields that $\|\widetilde{U}\mathbf{1}\| = |\alpha|\|e_1\|$, i.e., $|\alpha| = \|\widetilde{Q}\mathbf{1}\| = \|\mathbf{1}\| = \sqrt{M}$. Therefore, $Q - \frac{1}{n}\mathbf{1}\mathbf{1}^\top = Q^\top\left(D - \frac{1}{M}(Q\mathbf{1})(\widetilde{Q}\mathbf{1})^\top\right)\widetilde{Q}$ where $D - \frac{1}{M}(Q\mathbf{1})(\widetilde{Q}\mathbf{1})^\top = D - \frac{\alpha^2}{M}e_1 e_1^\top = D - e_1 e_1^\top = \text{diag}(0, \sigma_2, \sigma_3, \ldots, \sigma_M)$, which proves that $\sigma_2$ is the largest singular value of $U - \frac{1}{n}\mathbf{1}\mathbf{1}^\top$, i.e., $\sigma = \sigma_2$.

Then, substituting $\sigma = \sigma_2$, $Ux - \mathbf{1}\overline{x} = (U - \frac{1}{M}\mathbf{1}\mathbf{1}^\top)x = U\Delta x$ (the last step follows from the item 1 of this Lemma) and $x - \mathbf{1}\overline{x} = (I - \frac{1}{M}\mathbf{1}\mathbf{1}^\top)x = \Delta x$ into eq. (50) yields that

$$\|U\Delta x\| \leq \sigma_2\|\Delta x\|. \tag{51}$$

A simple induction based on the above equality proves that $\|U^n \Delta x\| \le \sigma_2^n \|\Delta x\|$ for any $n \in \mathbb{N}^+$. Therefore, for any matrix $H = [h_1, \ldots, h_M] \in \mathbb{R}^{M \times M}$, we can prove that

$$\|W^n \Delta H\|_F = \sqrt{\sum_{m=1}^{M} \|W^n \Delta h_m\|^2} \le \sqrt{\sum_{m=1}^{M} (\sigma_2^n \|\Delta h_m\|)^2} = \sigma_2^n \|\Delta H\|_F.$$

$\square$

Based on Lemma C.3, we obtain the following inexactness of importance sampling ratio estimation $\widehat{\rho}_t^{(m)} \approx \rho_t$.

**Corollary 2.** *Under Assumption 4 and choosing $L \ge \mathcal{O}(\frac{\ln M + M \ln \rho_{\max}}{\ln \sigma_2^{-1}})$, the estimation error of the inexact global importance sampling ratio $\widehat{\rho}_i^{(m)}$ satisfies $\sum_{m=1}^{M} \left(\widehat{\rho}_i^{(m)} - \rho_i\right)^2 \le \sigma_2^{L/4}$. Therefore, the following inequalities hold.*

$$\sum_{m=1}^{M} \left\|A_i^{(m)} - A_i\right\|_F^2, \sum_{m=1}^{M} \left\|\widetilde{A}_t^{(m)} - \widetilde{A}_t\right\|_F^2 \le 4\sigma_2^{L/4} \tag{52}$$

$$\sum_{m=1}^{M} \left\|B_i^{(m)} - B_i\right\|_F^2, \sum_{m=1}^{M} \left\|\widetilde{B}_t^{(m)} - \widetilde{B}_t\right\|_F^2 \le \sigma_2^{L/4}, \tag{53}$$

$$\sum_{m=1}^{M} \left\|\widehat{b}_i^{(m)} - b_i^{(m)}\right\|^2, \sum_{m=1}^{M} \left\|\widehat{\widetilde{b}}_t^{(m)} - \widetilde{b}_t^{(m)}\right\|^2 \le \sigma_2^{L/4} R_{\max}^2. \tag{54}$$

*As a result, the following upper bounds hold.*

$$\|A_i^{(m)}\|_F, \left\|\widetilde{A}_t^{(m)}\right\|_F \le 2\rho_{\max} + 2 \tag{55}$$

$$\|B_i^{(m)}\|_F, \left\|\widetilde{B}_t^{(m)}\right\|_F \le \rho_{\max} + 1 \tag{56}$$

$$\|\widehat{b}_i^{(m)}\|, \|\widehat{\widetilde{b}}_t^{(m)}\| \le R_{\max}(\rho_{\max} + 1) \tag{57}$$

*Proof.* Eq. (5) can be rewritten into the following matrix form.

$$\left[\widetilde{\rho}_{i,L}^{(1)}; \ldots; \widetilde{\rho}_{i,L}^{(M)}\right] = U^L \left[\widetilde{\rho}_{i,0}^{(1)}; \ldots; \widetilde{\rho}_{i,0}^{(M)}\right].$$

Hence, the item 1 of Lemma C.3 yields that

$$\Delta\left[\widetilde{\rho}_{i,L}^{(1)}; \ldots; \widetilde{\rho}_{i,L}^{(M)}\right] = U^L \Delta\left[\widetilde{\rho}_{i,0}^{(1)}; \ldots; \widetilde{\rho}_{i,0}^{(M)}\right].$$

Then the item 2 of Lemma C.3 yields that

$$\left\|\Delta\left[\widetilde{\rho}_{i,L}^{(1)}; \ldots; \widetilde{\rho}_{i,L}^{(M)}\right]\right\|^2 \le \sigma_2^{2L} \left\|\Delta\left[\widetilde{\rho}_{i,0}^{(1)}; \ldots; \widetilde{\rho}_{i,0}^{(M)}\right]\right\|^2. \tag{58}$$

Denote $\rho_{\min} := \min_{m \in \mathcal{M}} \rho_i^{(m)}$. Then Assumption 4 implies that $\widetilde{\rho}_{i,0}^{(m)} = \ln \rho_i^{(m)} \in [\ln \rho_{\min}, \ln \rho_{\max}]$. Then it can be proved by iterating eq. (5) that $\widetilde{\rho}_{i,L}^{(m)} \in [\ln \rho_{\min}, \ln \rho_{\max}]$. Hence,

$$\frac{1}{M} \ln \rho_i = \frac{1}{M} \sum_{m=1}^{M} \ln \rho_i^{(m)} \in [\ln \rho_{\min}, \ln \rho_{\max}] \tag{59}$$

Then eqs. (58) and (59) imply that

$$\sum_{m=1}^{M} \left(\widetilde{\rho}_{i,L}^{(m)} - \frac{1}{M} \ln \rho_i\right)^2 \le \sigma_2^{2L} \sum_{m=1}^{M} \left(\widetilde{\rho}_{i,0}^{(m)} - \frac{1}{M} \ln \rho_i\right)^2 \le M \sigma_2^{2L} \ln^2(\rho_{\max}/\rho_{\min}). \tag{60}$$

Hence,

$$\left|\widetilde{\rho}_{i,L}^{(m)} - \frac{1}{M}\ln\rho_i\right| \le \sqrt{M}\sigma_2^L \ln\left(\frac{\rho_{\max}}{\rho_{\min}}\right) \stackrel{(i)}{\le} \frac{1}{2M}\ln\left(\frac{\rho_{\max}}{\rho_{\min}}\right), \tag{61}$$

where (i) uses the conditions that $L \ge \frac{12\ln M + (8M+10)\ln\rho_{\max}}{\ln(\sigma_2^{-1})}$ and $\sigma_2 \in [0,1)$.

Hence, eqs. (59) and (60) imply that

$$\widetilde{\rho}_{i,L}^{(m)} \le \ln\rho_{\max} + \frac{1}{2M}\ln\left(\rho_{\max}/\rho_{\min}\right). \tag{62}$$

Therefore, we obtain that

$$
\begin{aligned}
\sum_{m=1}^{M}\left(\widehat{\rho}_i^{(m)} - \rho_i\right)^2 &\stackrel{(i)}{=} \sum_{m=1}^{M}\left(e^{M\widetilde{\rho}_{i,L}^{(m)}} - e^{\ln\rho_i}\right)^2 \\
&\stackrel{(ii)}{\le} \sum_{m=1}^{M}\left[\max\left(e^{M\widetilde{\rho}_{i,L}^{(m)}}, e^{\ln\rho_i}\right)\right]^2\left(M\widetilde{\rho}_{i,L}^{(m)} - \ln\rho_i\right)^2 \\
&\stackrel{(iii)}{\le} M^3\sigma_2^{2L}\rho_{\max}^M\sqrt{\rho_{\max}/\rho_{\min}}\ln^2(\rho_{\max}/\rho_{\min}) \\
&\stackrel{(iv)}{\le} M^3\sigma_2^{2L}(\rho_{\max}^{M+2.5}/\rho_{\min}^{2.5}),
\end{aligned}
\tag{63}
$$

where (i) uses eq. (6), (ii) uses the Lagrange's Mean Value Theorem, (iii) uses eqs. (59), (60) and (62), (iv) uses the inequality that $\ln x < x$ for $x = \rho_{\max}/\rho_{\min} \ge 1$.

Since at least one of $\{\widetilde{\rho}_{i,0}^{(m)}\}_{m\in\mathcal{M}}$ equals $\ln\rho_{\min}$, we have

$$\ln\rho_i = \sum_{m=1}^{M}\widetilde{\rho}_{i,0}^{(m)} \le \ln\rho_{\min} + (M-1)\ln\rho_{\max}. \tag{64}$$

Then, eqs. (61) and (64) imply that

$$\widetilde{\rho}_{i,L}^{(m)} \le \frac{1}{2M}\ln\rho_{\min} + \left(1 - \frac{1}{2M}\right)\ln\rho_{\max} \tag{65}$$

Hence, we conclude that

$$\sum_{m=1}^{M}\left(\widehat{\rho}_i^{(m)} - \rho_i\right)^2 \stackrel{(i)}{=} \sum_{m=1}^{M}\left(e^{M\widetilde{\rho}_{i,L}^{(m)}} - e^{\ln\rho_i}\right)^2 \le \sum_{m=1}^{M}\max\left(e^{2M\widetilde{\rho}_{i,L}^{(m)}}, e^{2\ln\rho_i}\right) \stackrel{(ii)}{\le} M\rho_{\min}\rho_{\max}^{2M-1} \tag{66}$$

where (i) uses eq. (6), (ii) uses eqs. (64) and (65).

When $\rho_{\min} \ge \sigma_2^{L/2}$, eq. (63) implies that $\sum_{m=1}^{M}\left(\widehat{\rho}_i^{(m)} - \rho_i\right)^2 \le M^3\rho_{\max}^{M+2.5}\sigma_2^{0.75L}$; When $\rho_{\min} < \sigma_2^{L/2} < 1$, eq. (66) implies that $\sum_{m=1}^{M}\left(\widehat{\rho}_i^{(m)} - \rho_i\right)^2 \le M\rho_{\max}^{2M-1}\sigma_2^{L/2}$. Both imply $\sum_{m=1}^{M}\left(\widehat{\rho}_i^{(m)} - \rho_i\right)^2 \le \sigma_2^{L/4}$ since $L \ge \frac{12\ln M + (8M+10)\ln\rho_{\max}}{\ln(\sigma_2^{-1})}$.

Then, eq. (52) can be proved as follows.

$$\sum_{m=1}^{M}\left\|A_i^{(m)} - A_i\right\|_F^2 \le \left\|\phi(s_i)[\gamma\phi(s_{i+1}) - \phi(s_i)]^\top\right\|_F^2 \sum_{m=1}^{M}(\widehat{\rho}_i^{(m)} - \rho_i)^2 \le (1+\gamma)^2\sigma_2^{L/4} \le 4\sigma_2^{L/4} \tag{67}$$

The above inequality implies that $\sum_{m=1}^{M}\left\|\widetilde{A}_t^{(m)} - \widetilde{A}_t\right\|_F^2 = \sum_{m=1}^{M}\left\|\frac{1}{N}\sum_{i=tN}^{(t+1)N-1}(A_i^{(m)} - A_i)\right\|_F^2 \le 4\sigma_2^{L/4}$, where $\le$ applies Jensen's inequality to the convex function $\|\cdot\|_F^2$. Eqs. (53) and (54) can be proved similarly.

Eq. (52) implies that $\left\|A_i^{(m)} - A_i\right\|_F, \left\|\widetilde{A}_t^{(m)} - \widetilde{A}_t\right\|_F \le 2$. Hence, eq. (55) can be proved using triangle inequality and eq. (39). Eqs. (56) and (57) can be proved similarly. $\qquad\square$

The proof of Corollary 2 introduces a new technique, which includes discussion of two cases: $\rho_{\min} := \min_{m \in \mathcal{M}} \rho_i^{(m)}$ lies in $[\sigma^{L/2}, \rho_{\max}]$ and $(0, \sigma^{L/2}]$. This is necessary as the local average is applied to $\ln \widehat{\rho}_i^{(m)}$, which may be a large negative number that cannot ensure a small consensus error for a fixed number of local average steps $L$.

**Lemma C.4.** *Under the update rules of Algorithm 1 and choosing* $L \geq \frac{12 \ln M + (8M+10) \ln \rho_{\max}}{\ln(\sigma_2^{-1})}$, $\alpha \leq \frac{\beta}{2\rho_{\max}+2}$, *the parameters have the following upper bound.*

$$\max_{m \in \mathcal{M}} \|\theta_T^{(m)}\| + \max_{m \in \mathcal{M}} \|w_T^{(m)}\| \leq 2 \max_{m \in \mathcal{M}} (\|\theta_0^{(m)}\| + \|w_0^{(m)}\| + R_{\max})[1 + \beta(2\rho_{\max}+3)]^T. \tag{68}$$

*Proof.* Since $L \geq \frac{12 \ln M + (8M+10) \ln \rho_{\max}}{\ln(\sigma_2^{-1})}$, eqs. (55)-(57) hold. Hence, these equations and the update rule imply that $\|\theta_{t+1}^{(m)}\| \leq \sum_{m' \in \mathcal{N}_m} U_{m,m'} \|\theta_t^{(m')}\| + \alpha(\rho_{\max}+1)(2\|\theta_t^{(m)}\| + R_{\max} + \|w_t^{(m)}\|)$. Taking maximum with respect to $m$ yields that

$$\max_{m \in \mathcal{M}} \|\theta_{t+1}^{(m)}\| \leq \max_{m \in \mathcal{M}} \sum_{m' \in \mathcal{N}_m} U_{m,m'} \max_{m'' \in \mathcal{M}} \|\theta_t^{(m'')}\| + \alpha(\rho_{\max}+1)\big(2 \max_{m \in \mathcal{M}} \|\theta_t^{(m)}\| + R_{\max} + \max_{m \in \mathcal{M}} \|w_t^{(m)}\|\big).$$

$$\leq \alpha(\rho_{\max}+1)\big(2 \max_{m \in \mathcal{M}} \|\theta_t^{(m)}\| + R_{\max} + \max_{m \in \mathcal{M}} \|w_t^{(m)}\|\big) + \max_{m \in \mathcal{M}} \|w_t^{(m)}\|. \tag{69}$$

Similarly, it can be obtained that

$$\max_{m \in \mathcal{M}} \|w_{t+1}^{(m)}\| \leq 2\beta(\rho_{\max}+1) \max_{m \in \mathcal{M}} \|\theta_t^{(m)}\| + (1+\beta) \max_{m \in \mathcal{M}} \|w_t^{(m)}\| + \beta R_{\max}(\rho_{\max}+1). \tag{70}$$

Adding up eqs. (69) and (70) yields that

$$\max_{m \in \mathcal{M}} \|\theta_{t+1}^{(m)}\| + \max_{m \in \mathcal{M}} \|w_{t+1}^{(m)}\|$$

$$\leq 2(\alpha+\beta)(\rho_{\max}+1) \max_{m \in \mathcal{M}} \|\theta_t^{(m)}\| + (\alpha\rho_{\max}+\alpha+\beta+1) \max_{m \in \mathcal{M}} \|w_t^{(m)}\| + R_{\max}(\alpha+\beta)(\rho_{\max}+1)$$

$$\overset{(i)}{\leq} \beta(2\rho_{\max}+3) \max_{m \in \mathcal{M}} \|\theta_t^{(m)}\| + (1.5\beta+1) \max_{m \in \mathcal{M}} \|w_t^{(m)}\| + 0.5\beta R_{\max}(2\rho_{\max}+3)$$

$$\leq [1 + \beta(2\rho_{\max}+3)]\big(\max_{m \in \mathcal{M}} \|\theta_t^{(m)}\| + \max_{m \in \mathcal{M}} \|w_t^{(m)}\|\big) + 0.5\beta R_{\max}(2\rho_{\max}+3),$$

where (i) uses the condition that $\alpha \leq \frac{\beta}{2\rho_{\max}+2}$ and (ii) uses $\rho_{\max} \geq 1$. By iterating the inequality above and using $\max_{m \in \mathcal{M}} \|\theta_0^{(m)}\| + \max_{m \in \mathcal{M}} \|w_0^{(m)}\| \leq 2 \max_{m \in \mathcal{M}} (\|\theta_0^{(m)}\| + \|w_0^{(m)}\|)$, we prove eq. (68). $\qquad\square$

