# OpenReview forum: "Multi-Agent Off-Policy TDC with Near-Optimal Sample and Communication Complexities"
_TMLR — Accepted by TMLR_

### Review · Reviewer_TD45 · 2022-04-27

**Summary Of Contributions:**

The paper proposes a new decentralized off-policy method for the multi-agent policy evaluation problem under linear function approximation.
The novel method is derived from the two-time scale TDC method from single-agent problems with two phases. The first phase includes decentralized TDC with a novel method to efficiently compute an approximation of the global importance sampling ratio while preserving communication restrictions across agents. Futhermore, the first phase requires mini-batch sampling, which can help reduce variance. The first phase is then followed by a second local averaging phase to guarantee consensus of the agents' parameters and prove convergence. The sample complexity of the proposed method is given and contrasted with related works. The method seems to be the first fully decentralized TDC method without making assumptions such as independent MDPs or policy sharing across agents.

**Broader Impact Concerns:**

I do not have any concerns regarding broader impact.

**Requested Changes:**

I propose the following changes that I think would improve the work.

## Critical changes
* Introduce all notations used within the main body, without exception, and to include missing definitions in the appendix. Please see my comment above.
* Discussion about the __requirement__ of a mini-bath size greater than 1 and why this is needed in practice or theory. If it is a conjectured to be a limitation of the analysis it would be helpful to include experiments with a batchsize of $N=1$.
* Address the aforementioned issue regarding Lemma C.3 above.

## Changes to strengthen the paper
* Resolving all issues regarding notation. I would also suggest to have an index of notation at the start of the appendix for __all__ the symbols. I think this would greatly improve the readability of the paper.
* Address clarity issues mentioned above. Mostly regarding the mini-batches and how it is used in conjunction with the global importance estimating procedure.
* Include the discussion of the choice of values in the matrix of V and how this may affect performance (see more details above).

**Strengths And Weaknesses:**

The main strength of the paper is the algorithmic contribution, with sample complexity analysis and being fully decentralized with less restrictive assumptions than previous works, makes for a potential strong contribution in the area. However, I find several issues in the presentation of the work regarding some important limitations that are not fully discussed but are only reflected in the proofs, as well as possible theoretical issues  and other more minor issues including clarity in algorithm description, and notation. I go into more detail below.

## Clarity in algorithm description an communication complexity

From the pseudo-code and algorithm description it is not clear how exactly the minibatches are utilized. I would assume that for a mini-batch of size $N$ and $L$ iterations of approximating $\hat{\rho}_i^{(m)}$ that there should be $N$ estimates of $\hat{\rho}_i^{(m)}$ (one for each sample in the mini-batch). Is this correct? If so, how is it possible for the M agents to produce one mini-batch update in parallel (as claimed in second for loop of the pseudo-code)? From my understanding, L rounds of equations (5-6) cannot be implemented in parallel, iteration $\ell \in \{0,\cdots,L-2\}$ needs to be completed (requiring information from all agents within the neighbourhood of agent $m$) before iteration $\ell +1$. Additionally, it is unclear how these approximation steps are reflected in communication complexity, shouldn't complexity increase as $L$ increases? These global importance sampling approximation steps don't seem to be reflected in the communication complexity. Additionally, it is unclear what is exactly meant by communication complexity it would be useful to formally define it in the paper.

## Limitations of algorithm and Theorem 1

The presentation of theorem 1 in the main body suggests a fast linear convergence result, however it does not clearly state stepsize or mini-batch conditions which is common in other works. See for example the closely related work of Sun et al. (2020) (Theorem 1) where the stepsize condition is clearly stated without using big O notation. I believe the big O notation hides significant restrictions that are not discussed in the paper. From looking at the proof it seems that the linear convergence comes at a significant cost with a __required__ mini-batch size greater than 1 and possibly small stepsizes. More specifically the minibatch condition (which is required for each step of phase 1) is

$N \geq \max  \left( \frac{8 \nu \rho_\max}{\lambda_2(1-\delta)}, \frac{128 \rho_\max^2 (\nu+1)}{\lambda_1(1-\delta)}\left(1+\frac{\rho_\max}{\lambda_2}\right)^2, \frac{6408 \beta \rho_\max^2(\nu+1)}{\alpha \lambda_1 \lambda_2(1-\delta)}\left(1+\frac{1}{\lambda_2}^2\right)\right).$

This seems that $N$ cannot be taken to be a value of $1$ and may in fact be very large. This seems like an important limitation of the algorithm that needs to be discussed in the main body. For example, the proposed method may not work in cases where one cannot query the environment twice at the same state. From my understanding, a large benefit of off-policy methods is to learn with minimal control over the data collection procedure. For example, in some applications I would imagine that it would be difficult or impossible to sample the behavioural policy more than once at a given state. Additionally, the potentially large mini-batch requirement suggests that the mini-batches are offering more than just variance reduction if convergence is not possible for $N=1$, indeed the experiments only show for the case $N \geq 10$. Is convergence observed in practice for $N=1$?

Similarly to the minibatch condition, the stepsizes condition is not clearly stated and may require significantly small stepsizes which should be better stated in the theorem and discussed in the main body.

## Possible issue with Lemma C.3

The proof of the second item in Lemma C.3 is claimed to be given in the work of Qu & Li (2017), however, from my understanding the result in Qu & Li (2017) uses the __largest__ singular value while Lemma C.3 uses $\sigma_2$ which is the __second largest__ singular value. Therefore, it is unclear how the result from Qu & Li applies. I would recommend including the full arguments in the paper.

## Notation

The current notation and lack of definitions makes the paper difficult to follow, especially the proofs. Below I outline some possible issues with notation.

1. The notation used to describe the average updates (eq 12-13) should be defined if used in the main body. Furthermore, the average notation definitions are difficult to and are counter intuitive. For example, bar notation such as $\bar{b}_t$ is used to sometimes denote average over the $M$ agents but sometimes it is used to denote the average over the mini-batch (such as in $\bar{A}_t^{(m)}$). It would be much easier to follow with a more consistent stystem of notation.
2. $\rho^{(m)}(s,a)$ used in assumption 4 does not seem to be defined.
3. I assume $||A||_F$ denotes the frobenious norm but it is never formally introduced as such. It may also be helpful to state the fact that $||A x||_2 \leq ||A||_F ||x||_2$ which is often used in the proofs but never stated.
4. I assume $\lambda_\max(A)$ denotes the largest eigen value of $A$ but this notation is never introduced.
5. The following values used at the end of theorem 1 seem to never be defined: $c_{10},c_{11}, c_{8},c_{var},c_{sd}$. Is is mentioned that some of these are defined in appendix A but I cannot seem to find them.
6. What is the significance of $w_t^\ast$? It would be helpful to understand what this target parameter means. At the moment, the "optimization error" $\mathbb{E}[||w_{t+1}-w_t^\ast||^2]$ does not carry any meaning without further discussion.

## Other questions and comments

1. Do the specific values within the matrix $V$ carry any meaning? In Sun et al. I believe it is mentioned that the values of $V$ can be used as a design choice to optimize the value $\sigma_2$, is it the similar case here? For, example is there a reason for specific choices for the diagonal of $V$ used in the experiments? Why is the diagonal non-zero and why do the values differ across application? Is the matrix $V$ an important tuning parameter (the weights for the neighbors of each agent)?
2. It is mentioned that the analysis "establishes tight bounds of the consensus error cause by synchronizing the global importance sampling ratio, [...] where samples are correlated," I fail to see how samples are correlated when estimating the global sampling ratio, aren't the mini-bath samples i.i.d.? It is possible I am misunderstanding this sentence.
3. I think there is a typo in equation (3), shouldn't there only be subcript $i$ for $\rho$ and not a $t$ subscript?

# References

Sun, J., Wang, G., Giannakis, G. B., Yang, Q., & Yang, Z. (2020, June). Finite-time analysis of decentralized temporal-difference learning with linear function approximation. In International Conference on Artificial Intelligence and Statistics (pp. 4485-4495). PMLR.

Qu, Guannan, and Na Li. "Harnessing smoothness to accelerate distributed optimization." IEEE Transactions on Control of Network Systems 5.3 (2017): 1245-1260.

---

> ### Author Response · Authors · 2022-05-29
> **Authors' reply to Reviewer TD45**
>
> Thank you very much for reviewing our manuscript and providing valuable feedback.  Below is a response to the review comments. We have submitted a revised version with all revisions marked in `red`. Please let us know if further clarifications are needed.
>
> **Q1:** Clarity in algorithm description and communication complexity: Are there $N$ estimates of $\widehat{\rho}_i^{(m)}$ (one for each sample in the mini-batch)? How can agents produce mini-batch update of $\widehat{\rho}_i^{(m)}$ for $L$ rounds in parallel? What is communication complexity? Why does communication complexity not involve these $L$ rounds?
>
> **A:** Great questions.
>
> Yes, there are $N$ estimates of $\widehat{\rho}_i^{(m)}$ in each mini-batch.
>
> The $L$ rounds of update in eqs.(5), (6) are implemented sequentially, i.e., each round $\ell$ can be implemented by the agents in parallel, but the next round $\ell+1$ comes after round $\ell$. We have corrected this in our revised Algorithm 1. Thanks for pointing this out.
>
> We have defined communication complexity as the number of communication rounds in the introduction. We have repeated this definition in Section 4 of our revision.
>
> Theorem 1 states the communication complexity for synchronizing the model parameters $\theta_t^{(m)}$, which does not cover the communication complexity for updating the importance sampling ratio $\widehat{\rho}_i^{(m)}$. This is because the dimension of the model parameters is usually much larger than the scalar-valued importance sampling ratios. In the revision, we have added the communication complexity $TL$ for synchronizing $\widehat{\rho}_i^{(m)}$.
>
> **Q2:** Limitations of algorithm and Theorem 1: Big-O notations in the batchsize and stepsizes are vague, while Sun et al. (2020) did not use big-O notation. Also, big batchsize and possibly small stepsize are limitations of our algorithm and should be discussed in the main body. Does our algorithm converge in practice for $N=1$?
>
>
> **A: ** Good question. The full expressions of the convergence rates of Theorem 1 can be found in the proof of Theorem 1 in Appendix B, and we have indicated their locations in the revised Theorem 1. We use Big-O notation to highlight the dependence on the key parameters and simplify the presentation.
>
> Regarding the limitation of large batch size. Yes, our algorithm uses a large batchsize $N$, which effectively reduces the total number of iterations $T$ and therefore further reduces the communication complexity. At the same time, the overall sample complexity $NT$ is comparable to that of decentralized TD(0) with $N=1$. Therefore, it is beneficial to adopt mini-batch updates in theory. We also want to clarify that mini-batch updates can be implemented as easy as single-sample updates. Specifically, all we need is to follow the behavior policy $\pi_b$ and collect a trajectory of samples {$s_0, a_0^{(m)}, s_{1}, R_0^{(m)}, .. $}$_{m=1}^M$. Then, we group these trajectory of samples into mini-batches and directly perform the mini-batch updates in eqs. (7), (8). The data collection process is the same as the on-policy data collection process, and we do not need to query the mini-batch samples at the same state.
>
> Regarding the limitation of small stepsize, we agree that our stepsize $\alpha=\mathcal{O}\big(\frac{1-\sigma_2}{\sqrt{M}}\big)$ can be small when $\sigma_2\approx 1$ and $M$ is large, and we have mentioned this limitation in the revision. However, we note that our stepsize does not scale with the target accuracy $\epsilon$, whereas Sun et. al. (2020) requires the stepsize to scale with $\min\big[\mathcal{O}(\epsilon),\mathcal{O}\big(\sqrt{\frac{\epsilon}{M}}(1-\sigma_2)\big)\big]$ (since their convergence rate in proposition 2 is below $\epsilon$).
>
> We have added decentralized TDC with $N=1$ in our experiments and found that it can converge and has comparable sample complexity to that of decentralized TDC with other $N$ and that of decentralized TD(0).
>
> **Q3:** Possible issue with Lemma C.3.
>
> **A:** Good point. In Qu \& Li (2017), $\sigma$ is the spectral norm of $W-\frac{1}{n}\mathbf{1}\mathbf{1}^{\top}$, which actually equals to the second largest singular value of the communication matrix $W$ as claimed on their page 4.
>
> **Q4:** Define the notations in eqs. (12-13) in the main body. Also, bar notation denotes agents' average only in some cases.
>
> **A:** Thanks for the suggestion. We have clarified the notations in eqs. (12-13) in our revision.
> Regarding the bar notation, we use it only to denote agents' average in our revision. For example we use $\overline A_t:=(1/M)\sum_{m=1}^M A_i^{(m)}$ and $\widetilde A_t^{(m)}:=(1/N)\sum_{i=tN}^{(t+1)N-1}A_i^{(m)}$ to denote agents' average and minibatch average, respectively.
>
> **Q5:** $\rho^{(m)}(s,a^{(m)})$ in assumption 4 does not seem to be defined.
>
> **A:** Thanks for pointing this out. We have added the definition $\rho^{(m)}(s,a^{(m)}):=\frac{\pi^{(m)}(a^{(m)}|s)}{\pi_b^{(m)}(a^{(m)}|s)}$ in our revision.

---

> > ### Author Response · Authors · 2022-05-29
> > **Authors' reply to Reviewer TD45 (2)**
> >
> >
> > **Q6:** Define $||A||_F$ as the Frobenius norm and state the fact that $||Ax||_2\le||A||_F||x||_2$ in the proof.
> >
> > **A:** Thank you for the suggestion. We have clarified that in the revision.
> >
> > **Q7:** $\lambda_{\max}(A)$ is not defined. Is it the largest eigenvalue of $A$?
> >
> > **A:** Yes, it is the largest eigenvalue of $A$. We have added this definition in our revision. Thank you for pointing that out.
> >
> > **Q8:** $c_{10}$, $c_{11}$, $c_{8}$, $c_{\text{var}}$, $c_{\text{sd}}$ at the end of Theorem 1 are not defined.
> >
> > **A:** Good point. These notations are not needed and we have removed them in our revision. Sorry about the confusion.
> >
> > **Q9:** What is the meaning of $w_t^*$ and $\mathbb E[||w_{t+1}-w_t^*||^2]$?
> >
> > **A:** Great question. $w_t^*=-C^{-1}(A\overline{\theta}_t+b)$ is the optimal auxiliary parameter corresponding to $\overline{\theta}_t$. It is used to track the product of two expectation terms involved in the gradient of the MSPBE objective function, and is the key to develop the two-time scale updates of TDC. We have added this definition at the beginning of Section 4. Thank you for pointing this out.
> >
> > We agree that the term $\mathbb E[||w_{t+1}-w_t^*||^2]$ should not be understood as optimization error. It is an intermediate step to derive an upper bound of the term $\mathbb E[||w_{t+1}-w_{t+1}^*||^2]$, which corresponds to the optimization error of learning the optimal auxiliary parameter $w_{t+1}^*$ in iteration $t+1$. We have clarified this point in the revision.
> >
> > **Q10:** What do values within the matrix $V$ mean? Can they be used as a design choice to optimize $\sigma_2$? Why do we use nonzero diagnonals? Why do $V$ differ across applications? Is $V$ an important tuning parameter?
> >
> > **A:** $V_{ij}$ can be seen as the weighted average coefficient from agent $j$ to agent $i$. For example, if agent $i$ receives quantity $a_j$ from every other agent $j$, then it computes the weighted average $\sum_{j}V_{ij}a_j$ to update its local parameters.
> >
> > Yes, $V$ can be used as a design choice to optimize $\sigma_2$.
> >
> > In the experiments, we select diagonally dominant $V$ so that its $\sigma_2$ is bounded away from zero. This makes it easier to show the effect of $L$--the number of communications per iteration for synchronizing the importance sampling ratio $\widehat{\rho}_i^{(m)}$.
> > $V\in\mathbb{R}^{M\times M}$ differs among these applications because they have different number of agents $M$.
> >
> > $V$ is an important tuning parameter since its $\sigma_2$ affects the communication complexity and sample complexity.
> >
> > **Q11:** How samples are correlated when estimating the global sampling ratio? Aren't the mini-batch samples i.i.d.?
> >
> > **A:** Throughout this paper, we consider Markovian samples sampled from the underlying MDP by following a behavior policy $\pi_b$. This is a realistic setting in which the samples are non-i.i.d. and are correlated. Specifically, we follow the behavior policy $\pi_b$ and collect a trajectory of samples {$s_0, a_0^{(m)}, s_{1}, R_0^{(m)}, s_1...$}$_{m=1}^M$ from the MDP. Then, we group these samples into mini-batches and directly perform the mini-batch updates in eqs. (7), (8). This is why we need Assumption 1 to control the correlation of the Markovian samples in the analysis.
> >
> > **Q12:** Typo: eq. (3) should not contain $t$.
> >
> > **A:** Good point. In our revision, we have replaced $t$ with $i$.
> >
> > **Q13:**  Introduce all notations used within the main body, without exception, and to include missing definitions in the appendix.
> >
> > **A:** Great suggestion. We have done that in our revision.
> >
> > **Q14:** Have an index of notation at the start of the appendix for all the symbols.
> >
> > **A:** Great suggestion. We have included a table of these notations in Appendix A.
> >
> > **Q15:** Include the discussion of the choice of values in the matrix of V and how this may affect performance.
> >
> > **A:** Thanks for the great advice. We have included this discussion right after Theorem 1.

---

> > > ### Comment · Reviewer_TD45 · 2022-05-31
> > > **Acknowledgement of Changes**
> > >
> > > I thank the authors for all the changes and hard work. I think the clarity of the paper has been improved significantly with the changes and agree with the authors' responses. I still think there can be improved clarity regarding Lemma C.3. It is difficult to find the appropriate arguments in Qu & Li (2017), I cannot find any specific details about the second eigenvalue unless the authors are referring to the Arxiv version of Qu & Li, even then it is not clear. Nevertheless, I cannot map the correct page numbers as referenced in the main paper, I think for example the reference to page 3 in Qu & Li might mean page 2 (or page 160 in the IEEE proceedings), more specifically Section II B.? Furthermore, upon looking more carefully I see that Qu & Li cite Olshevky & Tsitsiklis (2009) for this result, maybe it is more appropriate to cite them?
> > >
> > > I still recommend that full details of Lemma C.3 be given or a better reference to the previous works, in both cases of course the appropriate works should be cited.
> > >
> > >
> > >
> > > # References
> > >
> > > Qu, Guannan, and Na Li. "Harnessing smoothness to accelerate distributed optimization." IEEE Transactions on Control of Network Systems 5.3 (2017): 1245-1260.
> > >
> > > Olshevsky, Alex, and John N. Tsitsiklis. "Convergence speed in distributed consensus and averaging." SIAM journal on control and optimization 48.1 (2009): 33-55.

---

> > > > ### Author Response · Authors · 2022-06-01
> > > > **Further clarification for Lemma C.3**
> > > >
> > > > Thanks for the feedback.
> > > >
> > > > We do realize that the item 2 of our Lemma C.3 and the claim in Qu \& Li (2017) needs more detailed explanation. We clarify it as follows. Specifically, Qu \& Li (2017) claims that the largest singular value of $W-\frac{1}{n}\mathbf{1}\mathbf{1}^{\top}$ equals the second largest singular value of $W$, but does not provide a formal proof. Olshevsky \& Tsitsiklis (2009) does not provide the proof of this claim, either. We provide below a short and self-contained proof, which has been included in the revised paper.
> > > >
> > > > **Proof:** Consider the singular value decomposition of the doubly stochastic matrix $W=U^{\top}D\widetilde{U}$, where matrices $U$, $\widetilde{U}$ are unitary and matrix $D=\text{diag}(1,\sigma_2,\sigma_3,\ldots,\sigma_M)$ is diagonal with $1>\sigma_2\ge \sigma_3\ge \sigma_M \ge 0$. Note that $\mathbf{1}=W\mathbf{1}=U^{\top}D\widetilde{U}\mathbf{1}$, which implies that $U\mathbf{1}=D\widetilde{U}\mathbf{1}$. Similarly, $\mathbf{1}=W^{\top}\mathbf{1}=\widetilde{U}^{\top}DU\mathbf{1}$, which implies that $\widetilde{U}\mathbf{1}=DU\mathbf{1}$.
> > > > Combining the above two results, we conclude that
> > > >
> > > > $U\mathbf{1} = D\widetilde{U}\mathbf{1} = DDU\mathbf{1} = D^2 U\mathbf{1}$,
> > > >
> > > > that is, $(I-D^2)U\mathbf{1}=\mathbf{0}$. Since $I-D^2$ is a diagonal matrix where the first diagonal entry is zero and the rest diagonal entries are strictly positive, it must hold that all the entries of $U\mathbf{1}$ are zero except for its first entry, i.e., $U\mathbf{1}=\alpha e_1$ where $\alpha\in\mathbb{R}$ is the first entry of $U\mathbf{1}$ and $e_1=(1,0,0,...,0)$ is a basis vector. Hence, we conclude that $\widetilde{U}\mathbf{1}=DU\mathbf{1}=\alpha De_1=\alpha e_1$. In summary, we have proved that $U\mathbf{1} = \widetilde{U}\mathbf{1} = \alpha e_1$. Taking the norm of both sides yields that $\|U\mathbf{1}\|=\|\widetilde{U}\mathbf{1}\|=|\alpha|\|e_1\|$, i.e., $|\alpha|=\|\widetilde{U}\mathbf{1}\|=\|\mathbf{1}\|=\sqrt{n}$.
> > > > Therefore, $W-\frac{1}{n}\mathbf{1}\mathbf{1}^{\top}=U^{\top}\big(D-\frac{1}{n}(U\mathbf{1})(\widetilde{U}\mathbf{1})^{\top}\big)\widetilde{U}$, where $D-\frac{1}{n}(U\mathbf{1})(\widetilde{U}\mathbf{1})^{\top}=D-\frac{\alpha^2}{n} e_1e_1^{\top}=D-e_1e_1^{\top}=\text{diag}(0,\sigma_2,\sigma_3,\ldots,\sigma_M)$, which proves that $\sigma_2$ is the largest singular value of $W-\frac{1}{n}\mathbf{1}\mathbf{1}^{\top}$.

---

### Review · Reviewer_M45B · 2022-04-29

**Summary Of Contributions:**

The paper proposes a novel algorithm for distributed off-policy multiagent policy evaluation adapting the TDC (TD with gradient correction) algorithm. Using local mini-batch and distributed estimation of the global importance sampling ratio the authors show their algorithm achieves a sample complexity which matches the centralized case with a communication complexity that greatly improves over decentralized TD. Furthermore, their algorithm avoids sharing policy information, which may be considered as sensitive information in certain applications. The theoretical findings are supported by preliminary numerical simulations in small environments.

**Broader Impact Concerns:**

The paper's contribution is more on the theoretical side for a generic multiagent policy evaluation problem. I do not have any concern to flag.

**Requested Changes:**

** Required changes/clarifications
- In order to have an easier way to contrast the sample and communication complexity of Alg.1 with previous literature, please include in Sect.2 the complexity of decentralized TD(0) and in Sect.2 the complexity of centralized TDC.
- From my understanding, in order to compute local estimates of the global importance sampling ratios, Eq.5 requires communicating the local estimates $\tilde \rho$ across agents according to the topology of the communication graph. Since Eq.5 is reiterated L times and this is executed at each of the T iterations of the first phase of the algorithm, I would expect the total communication complexity to amount to T*L. But this does not seem to be the case. Please clarify this point.
- At each iteration t, "each agent collects N Markovian samples". I guess this means that we simulate pi for N steps, but agents only store the actions they play. Is that correct?
- At the beginning of Sect.4, the matrices A_t, B_t, C_t are never really properly defined.
- The big-O notation used in the statement of Thm.1 is a bit inaccurate, since it is not clear what is the variables considered to the define it.
- $\bar\theta_t$ is never defined
- I would expect Eq.10 to reduce to the centralized TDC case for M=1.
- In Fig.1 it looks like TDC is less accurate/slower than TD in terms of sample complexity. I guess this is the price to pay for the minibatches, which on the one hand allows less frequent communication, but on the other hand makes updates lag behind wrt to TD. Is that the case?
- Distributed TD(0) as introduced in Eq.2 is on-policy, but in the experiments you run an off-policy version. Clarify what the algorithm is.
- In Fig.2 TDC(L=1) tends to diverge as the importance sampling ratios are not accurate enough. This doesn't seem to affect the consensus as shown in Fig.3. Is this expected?
- The notation is A.1 is very difficult to parse. I suggest to replace it by a table with an explanation of the meaning of each term and their role in the proof.
- While the authors provide a sketch of the proof in the main paper, the appendix is difficult to browse. The authors did a good job in detailing and explaining all steps accurately, but the bigger plan or what is actually "happening" in the proof is hardly explained. I suggest to use the proof sketch to structure App.A and guide the reader through the different steps of the proof.

** Suggestions for improvement
- "Markovian samples" is typically used in theoretical papers (in particular in the analysis of TD-like algorithm), but it may be less obvious to the broader community. Clarify the meaning and contrast it with the iid case.
- The undirected graph G=(M,E) is introduced but never really used. You may want to introduce the notation N_m already when introducing the graph.
- The use of $V_{m,,m'}$ for the communication matrix clashes with the notation of the value function.
- Define formally a "doubly stochastic matrix"
- When introducing decentralized TD(0) (Eq.2) specify how the samples are obtained (i.e., on-policy). Also, it is never really explained what agents communicate exactly. Make it more explicit.
- The stationary distribution $mu_b$ is never properly defined.
- In MSPBE you define $\Pi$ as a projection operator, please define.
- In Sect 3.1 the use of index i (in Eq.3) and t (in Eq.4) is confusing. Please clarify.
- In Sect.5, you write $theta^* = A^{-1}b$ is the optimal model parameter. It would be good to introduce this notion much earlier, for instance when introducing TDC.



**Strengths And Weaknesses:**

** Strengths
- Algorithmic contributions: While Alg1 (decentralized mini-batch TDC) is built around a simple adaptation of TDC to the distributed scenario, the algorithm contains a few crucial ingredients that are non-trivial (i.e., they were not used in previous works) and have clear impact on the guarantees of the algorithm. Notably, the use of local mini-batches allows reducing the communication frequency and eventually the overall communication complexity. Furthermore, the agents perform a local estimation of the importance sampling ratios that allow computing a very accurate estimate of the global importance sampling ratio, which guarantees that accuracy of the updates. Finally, the algorithm includes a final communication step to ensure a desirable level of consensus is reached among all agents.
- Theoretical contributions: The authors prove the effect of each algorithmic choice in deriving the final sample and communication complexity bounds. The results are interesting and clearly show the advantage wrt distributed TD.

** Weaknesses
- While the paper is overall well written, it would benefit from more details and clearer explanations in the review of existing results, the algorithmic structure and theoretical results (see below). This would make the paper more accessible and contributions stronger.
- Experiments are designed more as a validation of the theory than a proper empirical assessment of the quality of the algorithm.

---

> ### Author Response · Authors · 2022-05-29
> **Authors' reply to Reviewer M45B (1)**
>
> Thank you very much for reviewing our manuscript and providing valuable feedback.  Below is a response to the review comments. We have submitted a revised version with all revisions marked in `red`. Please let us know if further clarifications are needed.
>
> **Q1:** Include in Section 2 the complexity of decentralized TD(0) and in Section 3 the complexity of centralized TDC.
>
> **A:** Thanks for the suggestion. In our revision, we have included those existing complexities and compared them with the complexities of our decentralized TDC.
>
> **Q2:** Why is the total communication complexity not $TL$?
>
> **A:** Good question. The communication complexity stated in Theorem 1 is for synchronizing the model parameter $\theta_t^{(m)}$. The communication complexity $TL$ is for synchronizing the importance sampling ratio $\rho$, which is a scalar and is much cheaper than synchronizing the model parameters. We have added and clarified this complexity in the revision. Thank you for pointing that out.
>
> **Q3:** Do agents simulate policy $\pi$ for $N$ steps and only store the actions they play per iteration?
>
> **A:** It is correct that $\pi$ is simulated for $N$ steps per iteration, but the agents do not need to store the actions. Specifically, the agents just need to store the quantities $\theta_t^{(m)}$, $w_t^{(m)}$, $\widetilde\rho_{i,\ell}^{(m)} $, and the second summation (add the $i$-th term at the ($i-tN$)-th step) of eqs. (7) and (8) respectively.
>
> **Q4:** At the beginning of Section 4, the matrices $A_t, B_t, C_t$ are never really properly defined.
>
> **A:** We defined these quantities in eq. (3) where $\pi$ and $\pi_b$ are respectively the joint targets and joint behavior policies. We have clarified these notations at the beginning of Section 4 in our revision. Thank you for pointing that out.
>
> **Q5:** The big-O notation used in the statement of Theorem 1 is a bit inaccurate.
>
> **A:** Thank you for pointing this out. We compressed some constants/parameters (e.g., $\lambda_1$, $\lambda_2$, $\delta$, $\nu$, $R_{\max}$) in the convergence rate of Theorem 1 to highlight the other key parameters. The full convergence rates can be found in the proof of Theorem 1 in Appendix B. Specifically, full expressions of eqs. (10) and (11) can be found in the second last line of eqs. (28) and (34) respectively. The hyperparameter choices mentioned after eq. (11) are specified at the end of Appendix B. We have clarified these in the revised Theorem 1.
>
> **Q6:** $\overline{\theta}_t$ is not defined.
>
> **A:** Thank you for pointing this out.  $\overline\theta_t:=(1/M)\sum_{m=1}^M\theta_t^{(m)}$ is defined in Appendix A.1. We have also defined this in Section 4.
>
> **Q7:** Eq. (10) should reduce to the centralized TDC case for $M=1$.
>
> **A:** When $M=1$ (so $\sigma_2=0$), the second last line of eq. (28) (detailed version of eq. (10)) achieves the order of the convergence rate of the centralized TDC (see eq. (30) in Appendix A of [1]).
>
> [1] Xu, Tengyu, and Yingbin Liang. “Sample complexity bounds for two timescale value-based reinforcement learning algorithms.” International Conference on Artificial Intelligence and Statistics. PMLR, 2021.
>
> **Q8:** In Fig.1, TDC is slower than TD in terms of sample complexity, which is the price to pay for the minibatches and makes update lag behind, correct?
>
> **A:** Great comment. Yes, TD(0) can be more sample efficient in practice due to the use of single-sample update. Another possible reason is that TDC has two-time scale updates, and therefore it takes more samples to converge in both time scales.
>
> **Q9:** How can we run decentralized TD(0) for off-policy evaluation in our experiment?
>
> **A:** Great question. The original decentralized TD(0) algorithm is simply our eq. (7) with setting $w_t^{(m)}\equiv 0$ and $\rho_i\equiv 1$ in the definition of $A_i^{(m)}$, $B_i^{(m)}$ and $\widehat{b}_i^{(m)}$. To adapt to off-policy evaluation, we simply use the estimated global importance sampling ratio $\widehat{\rho}_i^{(m)}$ via  eqs. (5) and (6) in decentralized TD(0). We have clarified this in our revision. Thank you for pointing that out.
>
> **Q10:** TDC with $L=1$ diverges in Fig. 2. Why doesn't that affect its consensus error in Fig. 3?
>
> **A:** Great question. Actually, it does affect. To elaborate, the consensus error of $L=1$ in Fig. 3 is much larger than that of the other $L$ values.
>
> **Q11:** Write a table to include the notations in Appendix A.1 as well as their meanings and roles in the proof.
>
> **A:** Thank you for the great suggestion. We have done that in our revision.
>
> **Q12:** Use the proof sketch to structure Appendix B and guide the reader through the different steps of the proof.
>
> **A:** Thank you for your great suggestion. At the beginning of each step of the proof in Appendix B, we have added a bolded paragraph heading and a brief introduction to explain this step.

---

> > ### Author Response · Authors · 2022-05-29
> > **Authors' reply to Reviewer M45B (2)**
> >
> > **Q13:** Clarify the meaning of “Markovian samples”.
> >
> > **A:** Thank you for your great suggestion. We have defined it in the introduction of our revision.
> >
> > **Q14:** Introduce $\mathcal{N}_m$ when introducing the communication graph.
> >
> > **A:** Thank you for your great suggestion, we have done that in our revision.
> >
> > **Q15:** The communication matrix $V_{m,m'}$ clashes value function.
> >
> > **A:** Great point. We have changed it to $U_{m,m'}$ in our revision.
> >
> > **Q16:** Define formally a “doubly stochastic matrix”.
> >
> > **A:** Thank you for your great suggestion. “Doubly stochastic matrix” denotes a matrix where the entries of each row and those of each column sum up to 1. We have added this definition to Assumption 5 in our revision.
> >
> > **Q17:** When introducing decentralized TD(0) (eq. (2)), specify explicitly how the samples are obtained and what agents communicate.
> >
> > **A:** Great suggestion. we have done that in our revision.
> >
> > **Q18:** The stationary distribution $\mu_b$ is never properly defined.
> >
> > **A:** Thanks for pointing it out. We have made its definition more clear in Section 3.1 of our revision: $\mu_b$ is the state stationary distribution under the behavior policy $\pi_b$.
> >
> > **Q19:** In Section 3.1, define the projection operator $\Pi$ more explicitly in MSPBE.
> >
> > **A:** Thank you for your great suggestion. We have added the definition in our revision.
> >
> > **Q20:** In Section 3.1, the use of index $i$ in eq. (3) and $t$ in eq. (4) is confusing. Please clarify.
> >
> > **A:** Index $i$ means the $i$-th sample, and $t$ means the $t$-th update iteration.
> >
> > **Q21:** Introduce $\theta^*=A^{-1}b$ when introducing TDC.
> >
> > **A:** Thank you for your great suggestion. We have done that in our revision.

---

> > > ### Comment · Reviewer_M45B · 2022-06-13
> > > **Revised version**
> > >
> > > Thanks to the authors for the revised version of the paper. The notation, algorithm, and theorem are now much clearer.

---

### Review · Reviewer_nejn · 2022-05-01

**Summary Of Contributions:**

This paper develops a sample-efficient and communication-efficient decentralized TDC algorithm for multi-agent off-policy evaluation. The communication complexity and sample complexity are attained via theoretical analysis. Experimental results verify the theoretical results.

**Requested Changes:**

Comments:
1.	One of the main contributions of this paper is to establish the convergence of the off-policy TD algorithm. So, the authors are expected to provide more details to highlight the challenges in the off-policy setting compared with the on-policy counterpart. Moreover, if the intuition of how the challenges are overcome in the analysis is given, the contribution will be clearer to readers.
2.	 From the algorithmic perspective, it seems that there is nothing fundamentally new; thus, maybe the authors should focus more on exhibiting the theoretical analysis, at least, explaining the ideas behind the proof that did not appear in the existing literature.
3.	As for assumption 5, as long as V is doubly stochastic, it can be derived that the second largest singular value is in range [0, 1).
4.	The communication complexity does not take the communication topology into account. The total communications differ with different topologies even if number T+T’ is fixed. Densely connected network results more communications with the same number of iterations. The choice of communication matrix V also partially determines the convergence rate. The authors did not specify how V is chosen or optimized.
5.	For the figures where the x-axis represents sample complexity (resp. communication complexity), the reviewer suggests using number of samples (resp. number of communication rounds).
6.	The experiment settings are simple. The results are more convincing if more practical models are used.


**Strengths And Weaknesses:**

Strengths: the present paper gives theoretical analysis for off-policy TD learning, which is an essential contribution.
Weaknesses: The gap between off-policy and on-policy in theoretical analysis is not highlight. The idea in the analysis to address the challenges of off-policy is not clearly presented.

---

> ### Author Response · Authors · 2022-05-29
> **Authors' reply to Reviewer nejn**
>
> Thank you very much for reviewing our manuscript and providing valuable feedback.  Below is a response to the review comments. We have submitted a revised version with all revisions marked in `red`. Please let us know if further clarifications are needed.
>
> **Q1:** What is the challenge for analysis of off-policy evaluation compared with on-policy evaluation? How do we solve it?
>
> **A:** Great question. The major challenge in the analysis of off-policy evaluation is estimating the global importance sampling ratio and bounding the induced consensus error (see Corollary 2) in a tight way, as we elaborated in the second paragraph of Section 3.2 and the Step 2 of the proof sketch in Section 4. To briefly explain, to estimate the global importance sampling ratio, we propose to first estimate the average of the logarithm of the local importance sampling ratios, i.e., $\frac{1}{M}\sum_{m=1}^{M}\ln\rho_i^{(m)}$ (see the equation above eq.(5)). Due to the use of the logarithm function, a numerically small $\rho_i^{(m)}$ can lead to an exponentially large estimation error. Therefore, we need to develop special techniques to control the estimation error when the local ratios are indefinitely small. This helps establish the key estimation error bounds in Corollary 2, which are not needed in the on-policy evaluation case. We have highlighted this challenge in these two paragraphs in our revision.
>
> **Q2:** Are there new ideas behind the proof that did not appear in the existing literature?
>
> **A:** Similar to the answer to Q1, Step 2 of the proof sketch in Section 4 explains a new proof technique for developing tight bounds of the consensus error induced by synchronizing the global importance sampling ratio. To our knowledge, this is not seen in the existing literature on off-policy TD learning. We have clarified this in that paragraph in the revision.
>
> **Q3:** In Assumption 5, it is redundant to say that the second largest singular value $\sigma_2\in[0, 1)$ since $V$ is doubly stochastic.
>
> **A:** $V$ being doubly stochastic implies that $\sigma_2\in[0, 1]$. Our assumption assumes that $\sigma_2\in[0, 1)$, which rules out the case that $\sigma_2=1$ (e.g. identity matrix) and thus is not redundant.
>
> **Q4:** The communication complexity does not involve communication topology. How to choose or optimize $V$?
>
> **A:** Great question. Our communication complexity involves the second largest singular value $\sigma_2$, which is an important measure that reflects communication topology. For example, densely connected network tends to have smaller $\sigma_2$ than sparse network, and thus yields larger communication complexity and sample complexity.
>
> In practice, the selection of $V$ is often subject to the available communication topology. (For example, when only ring network is available, it requires $V_{ij}=0$ for any non-adjacent pair of agents $(i,j)$.) Therefore, it is preferable to select $V$ under the topological constraint with as small $\sigma_2$ as possible, which yields smaller communication complexity and sample complexity.
>
> We have added these points to our revision.
>
> **Q5:** Use “number of samples” and “number of communication rounds” in the x-axis of the figures.
>
> **A:** Good suggestion. We have updated that in our revision.
>
> **Q6:** Use practical models in the experiment.
>
> **A:** Thank you for the great suggestion. We have added a real-world experiment on the multi-agent path finding problem to Section 5.3. This problem is formulated in the widely-used grid-world environment. The experimental results are similar to those obtained in the simulated environment.

---

### Author Response · Authors · 2022-05-11
**Petition to extend the author response deadline**

Dear Reviewers and Editor,

Thanks a lot for providing valuable feedback to help improve the quality of our work. We are still working on the response letter and paper revision. However, it takes more time than we expected, and we have a conflict with the approaching NeurIPS conference deadline on May 19th. Given the situation, we sincerely petition for a two-week deadline extension. We will definitely try our best to submit the response on time.

Thanks a lot for your consideration,

Authors

---

### Decision · Action_Editors · 2022-06-14

**Recommendation:** Accept as is

**Comment:**

Much of the reviewers' concerns revolved around clarity in the presentation: including notation, claims in prior work, innovations needed to generalize prior work to the off-policy case, etc.  The authors did a thorough, and commendable, job of addressing these concerns and making considerable improvements to the manuscript!